# Typhoon rainstorm simulations with radar data assimilation in southeast coast of China

Jiyang Tian[1,2], Ronghua Liu[1,2], Liuqian Ding[1], Liang Guo[1,2], Bingyu Zhang[1,2]

[1] China Institute of Water Resources and Hydropower Research, Beijing, 100038, China

[2] Research Center on Flood & Drought Disaster Reduction of the Ministry of Water Resources, Beijing, 100038, China

*Correspondence to*: Ronghua Liu (liurh@iwhr.com)

**Abstract.** As an effective technique to improve the rainfall forecast, data assimilation plays an important role in meteorology and hydrology. The aim of this study is to explore the reasonable use of Doppler radar data assimilation to correct the initial and lateral boundary conditions of the Numerical Weather Prediction (NWP) systems. The Weather Research and Forecasting

(WRF) model is applied to simulate three typhoon storm events in southeast coast of China. Radar data from Changle Doppler radar station are assimilated with three-dimensional variational data assimilation (3-DVar) model. Nine assimilation modes are designed by three kinds of radar data and three assimilation time intervals. The rainfall simulations in a medium-scale catchment, Meixi, are evaluated by three indices including relative error (*RE*), critical success index (*CSI*) and root mean square error (*RMSE*). Assimilating radial velocity with time interval of 1 h can significantly improve the rainfall simulations

and outperforms the other modes for all the three storm events. Shortening the assimilation time interval can improve the rainfall simulations in most cases, while assimilating radar reflectivity always leads to worse simulation as the time interval shortens. The rainfall simulation can be improved by data assimilation as a whole, especially for the heavy rainfall with strong convection. The findings provide references for improving the typhoon rainfall forecasts in catchment scale and have great significance on typhoon rainstorm warning.

# 1 Introduction

Although the resolution of numerical weather prediction (NWP) system is increasing with the improvement of computational efficiency and abundance of observation data, the rainfall is still one of the most difficult meteorological factors to forecast (Lu et al., 2017; Avolio and Federico, 2018). Typhoon always comes along with heavy rainfall, which lead to great loss. However, due to the uncertainty of the rainfall and the imperfect generation of NWP systems, rainfall forecast with severe convection is unsatisfactory in medium and small catchment scale (Tian et al., 2017a). Data assimilation plays an important role in NWP and is always applied to correct the initial and lateral boundary condition of NWP system, which can effectively improve the rainfall forecast (Mohan et al., 2015; Liu et al., 2018).

Various kinds of observation data have been tested and assimilated by different assimilation methods. Wan and Xu (2011) simulated a heavy rainstorm using the Weather Research and Forecasting (WRF) model with Gridpoint Statistical Interpolation (GSI) data assimilation (DA) system in the central Guangdong Province of south-east China. The rainfall simulation error was reduced at 4-km grid scale by assimilating satellite radiance data, which helped to analyse rainfall causes accurately. Giannaros et al. (2016) evaluated a lightning data assimilation (LTNGDA) technique over eight rainfall events occurred in Greece. The verification score of the rainfall simulation were significantly improved by the employment of WRF-LTNGDA scheme, especially for heavy rainfall. Zhang et al. (2016) presented a regional ensemble data assimilation system that assimilated microwave radiances into WRF model for hydrological applications. The rainfall simulations were improved in terms of the accumulated rainfall and spatial rainfall distribution in the southeastern United States. Yucel et al. (2015) assimilated conventional meteorological data by three-dimensional variational data assimilation (3-DVar) model into the coupled atmospheric-hydrological system. The rainfall simulations as well as the runoff simulations are improved significantly at large basins in the western Black Sea Region of Turkey.

Due to their high spatiotemporal resolution, radar data are assimilated to correct the NWP system for mesoscale and microscale weather prediction (Milan et al., 2008; Zhao and Jin, 2008). Wang et al. (2013) tested the four-dimensional variational data assimilation (4-DVar) system by simulating a midlatitude squall-line case in the U.S. Great Plains, and the results indicated that radar data assimilation was able to improve rainfall forecasts from the WRF model at the convective scale. Liu et al. (2013) selected 4 storm events in a small catchment (135.2 km$^2$) located in southwest England to explore the effect of data assimilation for rainfall forecasts based on WRF model, and assimilating radar reflectivity by 3-DVar model can significantly improve the forecasting accuracy for the events with one-dimensional evenness in either space or time. By using the WRF model and Advanced Regional Prediction System (ARPS) 3-Dvar, Hou et al. (2015) improved the short-term forecast skill up to 9 hours by assimilating radar data in southern China.

Most studies focus on the assimilation algorithm and data selection. However, consistent conclusions have not been obtained for the option of radar reflectivity and radial velocity, and few studies pay attention on the time interval setting of data assimilation. Based on the WRF and 3-DVar model, Tian et al. (2017b) found that radar reflectivity assimilation led to better

rainfall simulation than radial velocity assimilation with the time interval of 6 h. Maiello et al. (2014) assimilated both radar reflectivity and radial velocity by 3-DVar model with 3 h assimilation cycle to improve the WRF high resolution initial condition, and the rainfall forecast became more accurate for several experiments in the urban area of Rome. Bauer et al. (2015) used the WRF model in combination with 3-DVar scheme to estimate the rainfall simulation, and the results showed that radar

data assimilation significantly improved the rainfall simulation by a 1-hour Rapid-Update Cycle with the high resolution of 3 km in Germany.

In reality, the operational forecast from meteorological department is guidance forecast with a large forecasting area. It is impossible to focus on the accuracy of the rainfall in small and medium catchment scale. Limited computing power makes that the number of restarting the forecasting system is only 2-4 times per day (Xie et al., 2016). The forecasting accuracy descends

gradually as the run time goes on, because the data assimilation is not in real time. Due to the poor accuracy in small scale and low-resolution, the rainfall forecasting from the meteorological department cannot be used directly as the input for hydrological forecasting in small and medium catchment (Tian et al., 2019). The local meteorological observations are necessary to be assimilated to improve the high resolution rainfall forecast. The NWP model maybe not corrected timely with long time interval of data assimilation, while shortening the time interval need a lot of computational resources and the observation errors in local

meteorological observations may be also amplified with high assimilation frequency in NWP model.

China suffers approximately nine tropical cyclones (TCs) each year in average (Shen et al., 2017). Most TCs develop into typhoons which always bring huge economic losses and a great number of casualties. Fujian is one of the most regularly affected provinces in the coastline of southeast China, and heavy rainfall caused by the interaction of typhoons and complex terrain lead to severe flood disasters. Accurate rainfall forecasting is of great significance to flood control and disaster

mitigation. However, typhoon rainstorms are still difficult to predict (Li et al., 2019). There are eight Doppler radar stations to obtain full coverage for meteorological monitoring in Fujian province. The plentiful radar data provide convenience and basis for the exploration of radar data assimilation in catchment scale.

In this study, Meixi catchment located in Fujian province is chosen as the study area. Due to the frequent heavy rainfall, the flood disasters are up to more than 20 times since 1949. On July 9, 2016, heavy rainfall caused by typhoon Nepartak leads to

severe flood, and attracted strong interest from the public, academics and government. Accurate rainfall simulation has a great practical significance in the study area. In order to explore the reasonable use of Doppler radar data assimilation to correct the initial and lateral boundary conditions of the NWP systems, the WRF model is applied to simulate three typhoon storm events affecting the Meixi catchment and 3-DVar model is used to assimilate the radar data to improve the typhoon rainstorm simulations. Nine assimilation modes are designed by three kinds of radar data (radar reflectivity, radial velocity, radar

reflectivity and radial velocity) and three assimilation time intervals (1h, 3h and 6h). The rainfall simulations are evaluated by three indices including relative error (*RE*), critical success index (*CSI*) and root mean square error (*RMSE*).

## 2. Study area and case studies

The Meixi catchment lies in east-central of Fujian province with subtropical monsoon climate (Fig.1). The drainage area is 956 km$^2$ and the average annual rainfall is approximately 1560 mm. There are eight rain gauges and one hydrologic station (Fig.2). In order to investigate the radar data assimilation effects on rainfall simulation, different kinds of rainfall processes caused by different stages of the typhoons are chosen in Meixi catchment. Three rainfall storms are shown in Table 1. Saola formed on July 28, 2012 and landed Fuding, Fujian until August 3. Then Saola weakened into a tropical storm at Jiangxi. With the movement of Saola, Meixi catchment was not directly affected, and the accumulated 24-h rainfall was only 84 mm. Hagibis landed Shantou, Guangdong on June 15, 2014 and then moved toward north with a fast-moving speed. Fortunately, Hagibis weakened into a tropical depression quickly during moving to northeastern Fujian on June 17. Event II occurred after the typhoon passed Meixi catchment and the accumulated 24-h rainfall was only 66 mm. Nepartak reached Fujian on July 9, 2016 and strengthened at Putian. Then Nepartak moved towards the northwest with a fast-moving speed and event III occurred when Nepartak was close to Meixi catchment. During the period, Nepartak reached its peak intensity. The 24 h accumulated rainfall was 242 mm, which led to high peak flow with 4710 m$^3$/s in Meixi catchment. The most destructive flood caused water and power cut-off in 11 villages and towns. Official figures standed at 74 dead and 15 missing from the flood, which also caused a direct economic loss of 5.234 billion yuan. Accurate rainfall forecasts appeared to be particularly important for Meixi catchment.

**[Figure 1 and 2, Table 1]**

## 3 Model description and numerical experiments

### 3.1 Model description

#### 3.1.1 WRF model and configurations

As the latest-generation mesoscale NWP system, the WRF model in version 4.0 is used to simulate the three typhoon storm events. Three nested domains with two-way nesting are designed and centred over Meixi catchment. The grid spacings are set at 4 km, 12 km and 36 km for the three nested domains from inside to outside (Chen et al., 2017). The grid numbers for the nested domain sizes are 100×100 for Dom 1, 210×210 for Dom 2 and 300×300 for Dom 3 (Fig. 1). Meixi catchment is completely covered by the innermost domain. All domains are comprised of 40 vertical pressure levels with the top level set as 50 hPa (Maiello et al., 2014). The NCEP Final (FNL) Operational Global Analysis data with 1°×1° grids are used to drive the WRF model and provide the initial and lateral boundary conditions. The time step is set to be 1 h for the WRF model output. The spin-up period of 12 h is applied to obtain a more accurate rainfall simulation. The option of physical parameterisations has a significant effect on the rainfall simulations, especially for microphysics, planetary boundary layer (PBL), radiation, land-surface model (LSM) and cumulus physics (Otieno et al., 2019). According to the previous studies on physics options selection, WRF Single-Moment 6 (WSM 6) for microphysics, Yonsei University (YSU) for PBL, Rapid

Radiative Transfer Model for application to GCMs (RRTMG) for longwave and shortwave radiation, Noah for LSM and Kain-Fritsch (KF) for cumulus physics are adopted in this study (Srivastava et al., 2015; Hazra et al., 2017; Cai et al., 2018; Tian et al., 2020).

**[Table 2]**

**3.1.2 3-DVar data assimilation and observation operator**

The fundamental of 3-DVar data assimilation is to produce an optimal estimate of the true atmospheric state by the iterative solution of a prescribed cost function (Ide et al., 1997):

$$J(\mathbf{x}) = \frac{1}{2}(\mathbf{x} - \mathbf{x}^b)^T \mathbf{B}^{-1}(\mathbf{x} - \mathbf{x}^b) + \frac{1}{2}(\mathbf{y} - \mathbf{y}^0)^T \mathbf{R}^{-1}(\mathbf{y} - \mathbf{y}^0) \tag{1}$$

where $\mathbf{x}$ is the vector of the analysis, $\mathbf{x}^b$ is the vector of first guess or background, $\mathbf{y}$ is the vector of the model-derived observation that is transformed from $\mathbf{x}$ by the observation operator $\mathbf{H}$, i.e., $\mathbf{y} = \mathbf{H}(\mathbf{x})$, and $\mathbf{y}^0$ is the vector of the observation. $\mathbf{B}$ is the background error covariance matrix, and $\mathbf{R}$ is the observational and representative error covariance matrix. Equation (1) shows that the 3-DVar is based on a multivariate incremental formulation. Velocity potential, total water mixing ratio, unbalanced pressure and stream function are all preconditioned control variables. Radial velocity has already been derived into component winds that are the same as the analysis variables, hence radial velocity can be assimilated directly by Eq. (1).

However, radar reflectivity assimilation needs additional forward operator that associates the model hydrometeors with the radar reflectivity. Due to the wide applicability, the matrix of CV3 is adopted in this study to simplify the data assimilation procedure (Meng and Zhang, 2008).

The observation operator $\mathbf{H}$ in Eq. (1) links the model variables to the observation variables. For radar reflectivity, the observation operator is shown as (Sun and Crook, 1997):

$$Z = 43.1 + 17.5\log(\rho q_r) \tag{2}$$

where $Z$ is the radar reflectivity in dBZ, $\rho$ is the density of air in kg m$^{-3}$, and $q_r$ is rainwater mixing ratio in g kg$^{-1}$. Equation (2) is derived by assuming a Marshall-Palmer raindrop size distribution and that the ice phases have no effect on reflectivity. For radial velocity, the model-derived radial velocity $V_r$ can be calculated as (Tian et al., 2017b):

$$V_r = u\frac{x - x_i}{r_i} + v\frac{y - y_i}{r_i} + (w - v_t)\frac{z - z_i}{r_i} \tag{3}$$

where ($u$, $v$ and $w$) is the three-dimensional wind field, ($x$, $y$, $z$) represents the location of the observation point and ($x_i$, $y_i$, $z_i$) represents the location of the radar station. $r_i$ is the distance between the location of a data point and the radar station, $v_t$ is the hydrometer fall speed or terminal velocity. According to Sun and Crook (1998), $v_t$ can be given by:

$$v_t = 5.40a(\rho q_r)^{0.125} \tag{4}$$

$$a = \left( p_0 \middle/ \overline{p} \right)^{0.4}$$ (5)

where $a$ is the correction factor, $\overline{p}$ is the base-state pressure and $p_0$ is the pressure at the ground.

## 3.2 Numerical experiments

### 3.2.1 Assimilation modes

An S-band Doppler weather radar located at Changle can completely cover Meixi catchment. The observation radius of Changle radar reaches 250 km and the distance between Meixi catchment and radar station is less than 100 km (Fig. 2), which makes the quality of radar data credible. The assimilated data, radar reflectivity and radial velocity, can be obtained once every 6 min continuously. All the radar data with quality control are provided by the newest generation weather radar network of
China (CINRAD/SC). The observation error standard deviations of radar reflectivity and radial velocity are set 2 dBZ and 1 m s$^{-1}$ in the 3-DVar model, respectively (Caya et al., 2005). The radar data assimilation modes are designed by three kinds of radar data (radar reflectivity, radial velocity, radar reflectivity and radial velocity) and three assimilation time intervals (1h, 3h and 6h). The rainfall simulation without data assimilation is used as control mode. Nine modes are shown in Table 3.

**[Figure 2 and Table 3]**

### 3.2.2 WRF cycling runs for data assimilation

In order to obtain the whole process of the rainfall simulation, the running time is set as 36 h, 42 h and 36 h for storm event I, II and III, respectively. As shown in Fig 3, the cycling runs are set according to the time interval of data assimilation and run 1 can be regarded as the WRF run without data assimilation. The dashed line segment represents the model spin-up. The first-
guess generated by run 1 is applied to drive run 2. As time progresses, the first guess file generated in the previous run is used to provide the initial conditions for the following run. For storm event I, data assimilation starts on 3 August 2012 at 00:00 and occurs with intervals of 6 h, 3 h and 1 h. The start time of data assimilation is 18:00 on 17 June 2014 and the end time is 00:00 on 18 June 2014 for storm event II. Data assimilation takes place on 8 July at 18:00 and ends on 9 July at 18:00 with intervals of 6 h, 3 h and 1 h for storm event III.

**[Figure 3]**

## 4 Rainfall evaluation statistics

In this study, the observation of areal rainfall in Meixi catchment is averaged by the 8 stations with Thiessen polygon method (Sivapalan and Blöschl, 1998), while the simulation of areal rainfall is averaged from all grids of the WRF model inside the Meixi catchment. The relative error (*RE*) is used to evaluate the total rainfall amount simulation:

$$RE = \frac{P' - P}{P} \times 100\% \tag{6}$$

where $P'$ is the simulation of 24-h accumulated areal rainfall, and $P$ is the observation of 24-h accumulated areal rainfall.

The spatiotemporal patterns of the rainfall simulation are evaluated by the critical success index (CSI) and modified root mean square error (m-RMSE), which is defined as the ratio of root mean square error (RMSE) to the mean values of the corresponding observations (Liu et al., 2012; Prakash et al., 2014; Agnihotri and Dimri, 2015):

$$CSI = \frac{1}{N} \sum_{i=1}^{N} \frac{NA_i}{NA_i + NB_i + NC_i} \tag{7}$$

$$m\text{-}RMSE = \frac{\sqrt{\frac{1}{M} \sum_{j=1}^{M} \left(P'_j - P_j\right)^2}}{\frac{1}{M} \sum_{j=1}^{M} P_j} \tag{8}$$

The CSI is calculated based on the rain or no rain contingency table (Dai et al., 2019). Table 4 shows that rainfall<0.1 mm/h as the threshold is regarded as no rain. In order to evaluate the simulation of spatial rainfall distribution by CSI, $NA_i$, $NB_i$, $NC_i$ at each time step $i$ ($i$=1 h) are calculated by comparing the rainfall observation with simulation extracted at 8 rain gauge locations, and then the values of $NA_i$, $NB_i$, $NC_i$ at all time steps are averaged to produce the final verification results. Therefore, $N$ refers to the total time steps ($N$=24). For temporal dimension evaluation, $NA_i$, $NB_i$, $NC_i$ are first calculated using the time series data of simulations and observations at each rain gauge $i$ ($i$= 1), then the averaged index values of all rain gauges are regarded as the final verification results. Thus instead of the simulation time steps, $N$ represents the total number of the rainfall gauges ($N$=8) for temporal dimension evaluation. The perfect score of CSI is 1.

**[Table 4]**

The m-RMSE is calculated using Eq. (8). For spatial dimension evaluation, $P'_j$ and $P_j$ refer to the simulation and observation of 24-h accumulated rainfall at rain gauge $j$, respectively. $M$ is the total number of rain gauges ($M$=8). For temporal dimension evaluation, $P'_j$ and $P_j$ are the simulation and observation of areal rainfall at each time $j$, respectively. $M$ represents the total number of time ($M$=24). The perfect score of RMSE is 0.

## 5 Results

### 5.1 Accumulated rainfall simulation of the nine data assimilation modes

Nine data assimilation modes for three storm events are evaluated by RE for 24-h accumulated areal rainfall. The average values of the RE (ARE) of three storm events for each mode are also calculated. As shown in Table 5, data assimilation modes make the rainfall simulation worse according to REs of event I. Only mode 8 has the closest rainfall simulation to the

observation in the nine data assimilation modes and the *RE* is below 1%. For event II, all data assimilation modes can improve the accumulated rainfall simulations, while for event III, most modes make the accumulated rainfall simulations better except for mode 2 and 3. Mode 8, i.e. assimilating radial velocity with time interval of 1 h, always has the lowest *RE* and performs the best. The improvement of the rainfall simulation is the most obvious for event III and the rainfall magnitude is quite close to the observation, which has important significance in torrential rainfall forecast and catastrophic flood forecast at medium and small basins.

**[Table 5]**

### 5.1.1 Evaluation of assimilating effects for the different kinds of radar data

The assimilating effects for three kinds of radar data are compared in different assimilating time intervals. Based on the *RE*s of mode 1, 2 and 3, assimilating radar reflectivity always leads to better simulations than assimilating other two kinds of radar data with time interval of 6 h. The worst mode for event I is assimilating both radar reflectivity and radial velocity while for event II and III is assimilating radial velocity. According to the *RE*s of mode 4, 5 and 6, assimilating radial velocity becomes the best choice with time interval of 3 h for the three storm events. Assimilating radar reflectivity has the worst performance in the three modes for event II and III, whereas assimilating radar reflectivity and radial velocity together performs the worst for event I. When the time interval of data assimilation becomes 1 h, the ranking of assimilation modes for accumulated rainfall simulation is assimilating radial velocity > assimilating radar reflectivity and radial velocity > assimilating radar reflectivity.

### 5.1.2 Evaluation of assimilation effects for the different assimilation time intervals

The influences of assimilating time intervals on rainfall simulation are analysed in this section. Comparing the *RE*s of mode 1, 4 and 7, shortening the time interval of radar reflectivity assimilation has no obvious improvement for rainfall simulation and even makes the rainfall simulation worse. For assimilating radial velocity, all the rainfall simulations of three storm events become more accuracy and the assimilation effects are significantly improved as the time interval shortens from 6 h to 1 h. The *RE*s of the three storm events are all lower than 8% for the radial velocity assimilation with time interval of 1 h. According to mode 3, 6 and 9, shortening the assimilation time interval can improve the rainfall simulations in most cases for assimilating radar reflectivity and radial velocity at the same time, while only the *RE* of mode 6 is higher than the *RE* of mode 3.

### 5.2 Spatiotemporal distribution of rainfall simulation based on the nine data assimilation modes

The spatiotemporal patterns of the rainfall have significant effect on flood peek and peak time in medium and small catchments. The indices of *CSI* and *RMSE* are also applied to compare the nine radar data assimilation modes. The average *CSI* values and the average *RMSE* values for the three storm events are also calculated for different modes.

### 5.2.1 Evaluation in spatial dimension

Table 6 indicates that although mode 8 with the highest *CSI* and lowest *RMSE* is the best choice in the nine data assimilation modes, rainfall simulations with data assimilation are always worse than without data assimilation for event I. Figure 4 shows that the observed rainfall centre locates in the east of Meixi catchment and it rains more in the upstream side than the

downstream side. However, the spatial distribution of the accumulated 24-h rainfall with no data assimilation is even. Mode 1 and 2 simulate the rainfall in the east side of Meixi catchment accurately, while the simulated rainfall in the west side is much smaller than the observation. The rainfall simulations of mode 3, 4, 5 and 6 are all even in spatial dimension and lower than the observation. The simulation of mode 7 shows that the rainfall in the downstream side is smaller than the upstream side, whereas the different distribution in east-west direction is not obvious and the simulated rainfall is smaller than the

observation. For mode 9, the spatial distribution of rainfall is also inconsistent with the observation. Only the simulation of mode 8 is close to the observed rainfall.

All *RMSEs* of the simulations with radar data assimilation are lower than without data assimilation, and only the *CSI* of mode 8 is higher than the simulation without data assimilation for event II. According to the rainfall distribution shown in Fig.5, the falling areas of simulated rainfall without data assimilation are totally wrong. The observed rainfall centre locates in the middle

of upstream and downstream catchment. However, the rainfall centres of mode1, 4 and 7 are all located in middle and lower region. The rainfall centre is in middle reaches for mode 2 while in western catchment for mode 3. For mode 5 and 6, the spatial distributions of rainfall are also inconsistent with the observation. The simulated rainfall in middle of upstream catchment is close to the observation for mode 8 and 9 but in the downstream catchment is poor. Although the spatial rainfall distributions have deviation, nine modes get better than the simulation without data assimilation as a whole.

Based on the Table 6, not all data assimilation modes help improve the rainfall simulation for event III. Mode 4, 5 and 9 have just a little improvement on rainfall simulation in spatial distribution, and only the simulation of mode 8 is closed to the observation. The simulations of mode 1, 2, 3, 4 and 5 are much lower than the observation in whole catchment. Most observed rainfall falls in the east of the catchment, while the simulated rainfall concentrates in the west for mode 6 and 9 and in the downstream catchment for mode 7.

<p align="center">**[Table 6 and Figure 4, 5, 6]**</p>

Based on the *CSI*s and *RMSE*s of mode 1, 2 and 3, assimilating radar reflectivity performs better than assimilating other two kinds of radar data in time interval of 6 h. Assimilating radar reflectivity and radial velocity at the same time always leads to the worst simulation. Comparing the two indices of mode 4, 5 and 6, assimilating radar reflectivity with time interval of 3 h can obtain the highest *CSI* for the three storm events, while assimilating radial velocity gets better performance than other two

modes based on *RMSE* for event I and II. With the time interval of 1 h, the ranking of assimilation mode for spatial distribution of rainfall simulation is assimilating radial velocity > assimilating radar reflectivity and radial velocity > assimilating radar reflectivity.

Comparing the two indices of mode 1, 4 and 7, rainfall simulations become even worse as the time interval of radar reflectivity assimilation shortens. For the three modes of assimilating radial velocity, most simulations become more accuracy and the assimilation effects are significantly improved as the time interval shortens from 6 h to 1 h. The same conclusion can be obtained for assimilating radar reflectivity and radial velocity at the same time, while the improvement is not as obvious as assimilating radial velocity.

### 5.2.2 Evaluation in temporal dimension

As shown in Table 7, the similar results can be found that most data assimilation modes cannot help the simulation of WRF model get better for event I. Only mode 8 is outstanding with the highest *CSI* and lowest *RMSE*. Figure 7 shows that the rainfall is concentrated at 10:00-16:00 for the observation. However, the main rainfall processes occur at 3:00-5:00 for mode 3, 4, 6, 3:00-5:00 and 14:00-15:00 for mode 1, 3:00-5:00 and 17:00-19:00 for mode 2, 3:00-5:00 and 9:00-10:00 for mode7, 3:00-5:00 and 20:00-24:00 for mode 9. The rainfall processes of Mode 5 and 8 are similar with the observation, while the rainfall simulations at 12:00-13:00 and 15:00 for mode 5 are worse than for mode 8.

According to the values of *CSI* and *RMSE*, only mode 8 and 9 are useful for the improvement of rainfall simulation and obvious improvement can be found in mode 8 for event II. The actual main rainfall process occurs at 15:00-18:00, while the time is advanced by 3 h for mode 1, 4, 5, 6 and 7. There is a delay of 3 h for main rainfall process of mode 3. Although the times of heavy rainfall for mode 2 and 9 are consistent with the observation, the areal rainfall at 18:00 is much higher than the observation.

For event III, although most *CSI*s of the simulation with radar data assimilation are lower than the simulation without data assimilation, the *RMSE*s show the opposite conclusions. From the Fig. 9, the observed rainfall is concentrated at 8:00-11:00. It can be easily found that mode 1, 2, 3, 4, 5, 6 cannot reproduce the heavy rainfall process in temporal dimension. The simulated rainfall is concentrated at 15:00-16:00 for mode 7. Only the simulation of mode 8 is basically consistent with the observation, while the simulation of mode 9 is worse than mode 8 at 8:00 and 10:00.

**[Table 7 and Figure 7, 8, 9]**

According to the *CSI*s and *RMSE*s of mode 1, 2 and 3, assimilating radar reflectivity with time interval of 6 h performs better than assimilating other two kinds of radar data. Assimilating radial velocity performs the worst for event II and assimilating radar reflectivity and radial velocity at the same time always leads to the worst simulation for event I and III. Based on the two indices of mode 4, 5 and 6, assimilating radial velocity with time interval of 3 h can obtain the highest *CSI* and lowest *RMSE* for the three storm events, while assimilating radar reflectivity and radial velocity at the same time performs worse than other two modes. For the time interval of 1 h, the ranking of assimilation modes for temporal distribution of rainfall simulation is assimilating radial velocity > assimilating radar reflectivity and radial velocity > assimilating radar reflectivity.

Comparing the indices of mode 1, 4 and 7, rainfall simulations for temporal distribution become even worse as the time interval of radar reflectivity assimilation shortens from 6 h to 1 h. For mode 2, 5 and 8, shortening the time interval can significantly improve the rainfall simulation by assimilating radial velocity. Mode 3, 6 and 9 indicate that rainfall simulation is improved

by shortening the time interval as a whole, whereas assimilating radar reflectivity and radial velocity at the same time with time interval of 3 h obtains the worst rainfall simulation for event II and III.

## 6 Discussion

In order to prove the accuracy of the assimilation results, typhoon paths for different assimilation modes are also simulated (Fig. 10). According to the simulations of Saola and Nepartak, the accurate typhoon path simulation always leads to accurate rainfall simulation. However, for typhoon Hagibis, the actual typhoon centre is far away from the Meixi catchment during the assimilation process. Hence, only the actual typhoon path for Hagibis is added in Fig. 10. Comparing the nine radar data assimilation modes, assimilating radial velocity with time interval of 6 h always performs the worst in rainfall simulation, while the rainfall simulation can be significantly improved by shortening the time interval of data assimilation. According to Eq. (3), although the physical process of the rainfall formation cannot be influenced by the radial velocity assimilation directly, the wind field and the water vapor transportation in initial and lateral boundary condition can be changed with the wind information in radial velocity. However, the wind field is quite variable especially for stormy weather. As the time interval becomes longer, the WRF model cannot be corrected by the radial velocity in time, whereas comparing with the simulation without data assimilation, the inevitable observation errors caused by atmospheric refractive in the radial velocity might lead to worse performance of the WRF model as the running time goes on (Montmerle and Faccani, 2009). That is the main reason that assimilating radial velocity with time interval of 6 h cannot obtain satisfactory simulations. Increasing the frequency of data assimilation, the effective information in radial velocity can correct the wind field and the water vapor transportation in the background field of WRF model timely, which is helpful to improve the rainfall simulation (Kawabata et al., 2014).

[Figure 10]

Wind field and water vapor transportation increment for different modes at the rainfall concentrating time are used to show how assimilating radial velocity and radar reflectivity affect the WRF model's initial and boundary conditions (Fig. 11-13). The shadows in Fig. 11-13 mean that water vapor transportation in analysis field is more than in background field. The darker the shadow in the figures, the more water vapor transportation increment, which is one of the most important factors that affects the amount of rainfall. For event I, anticlockwise wind field has contributed to the water vapor transportation from ocean to inland at 12:00 on August 3, 2012. Mode 1, 2, 5, 7, 8 and 9 all obtain the shadow area with obvious water vapor transportation increment. However, according to the coverage area of the shadow, only mode 5 and 8 influence Meixi catchment directly, which is consistent with the result that rainfall simulations with mode 5 and 8 are higher than simulations with no data assimilation at 12:00, while the increment of simulated rainfall is quite obviously for Mode 8. The differences of nine modes can be easily found in both wind field and water vapor transportation increment for event II. The water vapor transportation increases significantly in mode 2, 3, 8 and 9 at 18:00 on June 18, 2014, which has direct impact on the rainfall simulation in Meixi catchment. That is the main reason why simulated rainfalls with mode 2, 3, 8 and 9 are higher than the simulated rainfall with no data assimilation. However, the wind fields in these modes indicate a lack of warm and wet flow

supply and the rainfall weakens is almost inevitable after 18:00. Considering wind field and the range of water vapor transportation increment, the rainfalls may continue for a period time after 18:00 in mode 3, 8 and 9, which can also be reflected by hourly simulated rainfall shown in Fig. 8. For event III, the water vapor transportation increases significantly in mode 2, 3, 5, 6, 7, 8 and 9 at 6:00 on July 9, 2016, while only mode 8 and 9 affect Meixi catchment directly. The range of shadow in Fig.

13 is also consistent with the rainfall simulations. Wind field indicates that water vapor transportation is sufficient for mode 8 in a later time, which leads to a significant increase of simulated rainfall.

**[Figure 11, 12 and 13]**

Do further shortening the assimilation interval obtain better rainfall simulation? In terms of theory, the answer is yes, because improving the assimilation frequency can correct the initial and lateral boundary condition timely. However, the observation

errors of radial velocity may be amplified with high assimilation frequency in WRF model. There may be an "inverted u" relationship between accuracy of rainfall simulation and assimilation time interval (Myung et al., 2009). Further study should be carried out to investigate the optimal assimilation time interval.

On the contrary, assimilating radar reflectivity have little help for improving the rainfall simulation except for accumulated rainfall. Furthermore, rainfall simulation become even worse as the time interval of radar reflectivity assimilation shortens

from 6 h to 1 h. The background field of the WRF model has large difference with the actual weather situation, which can be reflected from the rainfall simulation without data assimilation against rainfall observation for the three storm events. As shown in Eq. (2), radar reflectivity is closely related to the humidity field and contains the information of rainfall hydrometeors (Wattrelot et al., 2014). That is to say the humidity information in radar reflectivity is quite different from the actual weather situation, which makes the 3-DVar data assimilation difficult to produce an optimal estimate of the true atmospheric state by

the iterative solution of a prescribed cost function. It should be also mentioned that due to the unchangeable, the matrix of CV3 has wide applicability but is not practical for all cases (Kong et al., 2017). The inadaptation of CV3 in the typhoon synoptic system and the large differences between the humidity information in radar reflectivity and the actual weather situation might be the main reasons for poor performance of radar reflectivity assimilation (Sun, 2005). The more frequent the radar reflectivity assimilation, the greater the pressure on the 3-DVar data assimilation model. Other data assimilation model with variable

background error covariance, such as the hybrid ensemble transform Kalman filter–three-dimensional variational data assimilation (ETKF-3DVAR) (Wang et al., 2012; Shen et al., 2016), should be tested for radar reflectivity assimilation in further study.

For the even rainfall events in space and time, such as storm event I, data assimilation should be used carefully. The WRF model has good performance on even rainfall simulation, especially for accumulated rainfall. The errors in the assimilated data

may have negative effect on the rainfall simulation. Additionally, assimilating other kinds of data and radar data together may help to improve the rainfall simulation. Though the conventional observations, such as upper-air and surface observations from meteorological station and sounding balloon, have low spatiotemporal resolution, the kinds of data are various and have wide coverage, which can help to improve the atmospheric motion in the WRF model at a large scale (Li et al., 2018). Yesubabu et al. (2016) indicates that assimilating the satellite observation also has positive effect on the rainfall simulation. Assimilating

different data sources together with the radar data may further improve the rainfall simulation in catchment scale. In reality, ECMWF is also tested for the data assimilation before FNL is used in this study (Zhang et al., 2018; Zhao et al., 2012). Although the rainfall simulations show some differences based on the two kinds of boundary conditions, the patterns of improvements from different data assimilation modes are quite similar and the same conclusions can be obtained.

## 7 Conclusion

Data assimilation is an efficient technique for improving the rainfall simulation. In order to explore the reasonable use of Doppler radar data assimilation to correct the initial and lateral boundary conditions of the NWP systems, three typhoon storm events, including Saola, Hagibis and Nepartak, are chosen to be simulated by WRF model with the nine modes in Meixi catchment located in southeast coast of China. The FNL analysis data with 1°×1° grids are used to drive the WRF model, and radar data from Changle Doppler radar station are applied to correct the initial and lateral boundary conditions. Three evaluating indices *RE*, *CSI* and *RMSE* are used to evaluate the nine radar data assimilation modes, which are designed by three kinds of radar data (radar reflectivity, radial velocity, radar reflectivity and radial velocity) and three assimilation time intervals (1h, 3h and 6h).

Contrastive analyses of the nine modes are carried out from three aspects: accumulated rainfall simulation, spatial rainfall distribution and temporal rainfall distribution. Four main conclusions are obtained: (1) in the nine radar data assimilation modes, assimilating radial velocity with time interval of 1 h can significantly improve the rainfall simulations and outperforms the other modes for all the three storm events; (2) shortening the assimilation time interval can improve the rainfall simulations in most cases, while assimilating radar reflectivity always leads to worse simulation as the time interval shortens; (3) radar reflectivity is the best choice for the data assimilation with time interval of 6 h, while radial velocity performs best for the data assimilation with time interval of 1 h; (4) data assimilation can improve the rainfall simulation as a whole, especially for the heavy rainfall with strong convection, whereas the improvement for even distributed rainfall in space and time is limited. More numerical simulation experiments should be tested in other catchments at different climate conditions. Further studies also should be carried out to investigate the data assimilation techniques to improve the simulation ability of heavy rainfall in the study areas.

## Author Contributions

All the authors contributed to the conception and the development of this manuscript. Jiyang Tian and Ronghua Liu contributed to radar data assimilation and manuscript writing. Liuqian Ding and Liang Guo assisted in the data assimilation modes design and analyses. Bingyu Zhang helped with the figure production.

**Acknowledgements**

This study was supported by the National Key R&D Program (2019YFC1510605, 2018YFC1508105), the National Natural Science Foundation of China (Grant No. 51909274, 51822906), and IWHR Research&Development Support Program (JZ0145B032020).

**Competing interests:** The authors declare that they have no conflict of interest.

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

**Table captions**

Table 1: Three storm events occurred in Meixi catchment.

Table 2. Physical parameterizations used in the WRF model.

Table 3: Radar data assimilation modes designed with data types and time intervals.

Table 4: Rain-no rain contingency table for the WRF simulation against observation.

Table 5: Accumulated rainfall simulation (mm) and RE values (%) of the nine data assimilation modes for three storm events.

Table 6: *CSI* and *RMSE* for spatial distribution of rainfall simulation based on the nine data assimilation modes.

Table 7: *CSI* and *RMSE* for temporal distribution of rainfall simulation based on the nine data assimilation modes.

**Figure captions**

Figure 1: The location of the Meixi catchment and three nested domains.

Figure 2: Radar scan area and Meixi basin.

Figure 3: The time bars of the assimilation cycling runs for (a) storm event I, (b) storm event II and (c) storm event III.

Figure 4: Spatial distribution of the simulated 24 h rainfall accumulations with nine data assimilation modes for Event I.

Figure 5. Spatial distribution of the simulated 24 h rainfall accumulations with nine data assimilation modes for Event II.

Figure 6. Spatial distribution of the simulated 24 h rainfall accumulations with nine data assimilation modes for Event III.

Figure 7: Time series bars of observed and simulated areal rainfall with nine data assimilation modes and the rainfall observation for Event I.

Figure 8: Time series bars of observed and simulated areal rainfall with nine data assimilation modes and the rainfall observation for Event II.

Figure 9: Time series bars of observed and simulated areal rainfall with nine data assimilation modes and the rainfall observation for Event III.

Figure 10: Typhoon path and simulations for (a) Saola, (b) Hagibis and (c) Nepartak.

Figure 11: Wind field and water vapor transportation increment (850hPa) for Event I at 12:00 on August 3, 2012.

Figure 12: Wind field and water vapor transportation increment (850hPa) for Event II at 18:00 on June 18, 2014.

Figure 13: Wind field and water vapor transportation increment (850hPa) for Event III at 6:00 on July 9, 2016.

**Table 1.** Three storm events occurred in Meixi catchment.

| Event ID | Typhoon | Storm start time (UTC+8) | Storm end time (UTC+8) | 24-h accumulated rainfall (mm) |
|----------|---------|--------------------------|------------------------|--------------------------------|
| I | Saola | 03/08/2012 00:00 | 04/08/2012 00:00 | 84 |
| II | Hagibis | 17/06/2014 21:00 | 18/06/2014 21:00 | 66 |
| III | Nepartak | 08/07/2016 18:00 | 09/07/2016 18:00 | 242 |

**Table 2**. Physical parameterizations used in the WRF model.

| Physical parameterization | Scheme |
| --- | --- |
| Microphysics | WRF Single-Moment 6 (WSM 6) |
| Planetary boundary layer (PBL) | Yonsei University (YSU) |
| Longwave and shortwave radiation | Rapid Radiative Transfer Model for application to GCMs (RRTMG) |
| Land-surface model (LSM) | Noah |
| Cumulus | Kain-Fritsch (KF) |

**Table 3.** Radar data assimilation modes designed with data types and time intervals.

| Modes | Time intervals of data assimilation | Assimilated radar data |
|---|---|---|
| 1 | 6 h | radar reflectivity |
| 2 | 6 h | radial velocity |
| 3 | 6 h | radar reflectivity and radial velocity |
| 4 | 3 h | radar reflectivity |
| 5 | 3 h | radial velocity |
| 6 | 3 h | radar reflectivity and radial velocity |
| 7 | 1 h | radar reflectivity |
| 8 | 1 h | radial velocity |
| 9 | 1 h | radar reflectivity and radial velocity |

**Table 4.** Rain-no rain contingency table for the WRF simulation against observation.

| Simulation/observation | Rain | No rain (<0.1 mm/h) |
|---|---|---|
| Rain | *NA* (hit) | *NB* (false alarm) |
| No rain (<0.1 mm/h) | *NC* (failure) | / |

**Table 5.** Accumulated rainfall simulation (mm) and *RE* values (%) of the nine data assimilation modes for three storm events.

| Modes | Event I | | Event II | | Event III | | *ARE* (%) |
|---|---|---|---|---|---|---|---|
| | Rainfall Simulation (mm) | *RE* (%) | Rainfall Simulation (mm) | *RE* (%) | Rainfall Simulation (mm) | *RE* (%) | |
| No radar data assimilation | 85.16 | 0.88 | 43.16 | 34.32 | 64.20 | 73.47 | 36.22 |
| 1 | 61.74 | 26.86 | 70.37 | 7.09 | 66.79 | 72.40 | 35.45 |
| 2 | 60.97 | 27.77 | 80.88 | 23.09 | 58.59 | 75.79 | 42.22 |
| 3 | 35.44 | 58.02 | 70.85 | 7.83 | 61.11 | 74.75 | 46.87 |
| 4 | 41.49 | 50.86 | 79.69 | 21.29 | 71.75 | 70.35 | 47.50 |
| 5 | 66.16 | 21.63 | 72.19 | 9.86 | 101.23 | 58.17 | 29.89 |
| 6 | 37.10 | 56.05 | 77.49 | 17.94 | 151.64 | 37.34 | 37.11 |
| 7 | 61.12 | 27.60 | 80.64 | 22.72 | 104.28 | 56.91 | 35.74 |
| 8 | 83.65 | 0.91 | 70.67 | 7.55 | 227.96 | 5.80 | 4.75 |
| 9 | 82.50 | 2.28 | 71.31 | 8.53 | 188.01 | 22.31 | 11.04 |

**Table 6.** *CSI* and *RMSE* for spatial distribution of rainfall simulation based on the nine data assimilation modes.

| Modes | Event I | | Event II | | Event III | | Average values for the three events | |
|---|---|---|---|---|---|---|---|---|
| | *CSI* | *RMSE* | *CSI* | *RMSE* | *CSI* | *RMSE* | *CSI* | *RMSE* |
| No radar data assimilation | 0.7368 | 0.1535 | 0.4479 | 0.5635 | 0.6146 | 0.7482 | 0.5998 | 0.4884 |
| 1 | 0.7614 | 0.4524 | 0.3587 | 0.4070 | 0.6154 | 0.7841 | 0.5785 | 0.5478 |
| 2 | 0.6925 | 0.4967 | 0.2829 | 0.4771 | 0.6154 | 0.7844 | 0.5303 | 0.5861 |
| 3 | 0.6865 | 0.6907 | 0.3346 | 0.4618 | 0.6154 | 0.8147 | 0.5455 | 0.6557 |
| 4 | 0.7436 | 0.6261 | 0.3561 | 0.4359 | 0.6154 | 0.7905 | 0.5717 | 0.6175 |
| 5 | 0.7358 | 0.5341 | 0.3195 | 0.4170 | 0.6096 | 0.7123 | 0.5550 | 0.5545 |
| 6 | 0.7143 | 0.6614 | 0.2212 | 0.4783 | 0.5909 | 0.6285 | 0.5088 | 0.5894 |
| 7 | 0.5337 | 0.4275 | 0.3949 | 0.4896 | 0.5938 | 0.8004 | 0.5075 | 0.5725 |
| 8 | 0.7395 | 0.1505 | 0.4504 | 0.3589 | 0.6287 | 0.1643 | 0.6062 | 0.2246 |
| 9 | 0.7368 | 0.4211 | 0.3168 | 0.3152 | 0.6146 | 0.4519 | 0.5561 | 0.3961 |

**Table 7.** *CSI* and *RMSE* for temporal distribution of rainfall simulation based on the nine data assimilation modes.

| Modes | Event I | | Event II | | Event III | | Average values for the three events | |
|---|---|---|---|---|---|---|---|---|
| | *CSI* | *RMSE* | *CSI* | *RMSE* | *CSI* | *RMSE* | *CSI* | *RMSE* |
| No radar data assimilation | 0.6875 | 0.6018 | 0.3718 | 1.3131 | 0.6146 | 1.9223 | 0.5580 | 1.2791 |
| 1 | 0.6830 | 1.0351 | 0.3069 | 1.3843 | 0.6034 | 1.8138 | 0.5311 | 1.4111 |
| 2 | 0.6458 | 1.1787 | 0.2483 | 2.0950 | 0.6034 | 1.8232 | 0.4992 | 1.6990 |
| 3 | 0.6421 | 1.2414 | 0.2969 | 1.4631 | 0.6034 | 1.8318 | 0.5141 | 1.5121 |
| 4 | 0.6674 | 1.1411 | 0.2902 | 2.2037 | 0.6034 | 1.8153 | 0.5203 | 1.7200 |
| 5 | 0.6796 | 1.1115 | 0.2969 | 2.0414 | 0.6145 | 1.7793 | 0.5303 | 1.6441 |
| 6 | 0.6667 | 1.2878 | 0.2031 | 2.6387 | 0.5851 | 2.0122 | 0.4850 | 1.9796 |
| 7 | 0.4549 | 1.5132 | 0.3125 | 2.3337 | 0.5938 | 1.8452 | 0.4537 | 1.8974 |
| 8 | 0.6877 | 0.3822 | 0.3969 | 0.7015 | 0.6221 | 0.8459 | 0.5689 | 0.6432 |
| 9 | 0.6875 | 1.3862 | 0.2663 | 1.1180 | 0.6146 | 1.1699 | 0.5228 | 1.2247 |

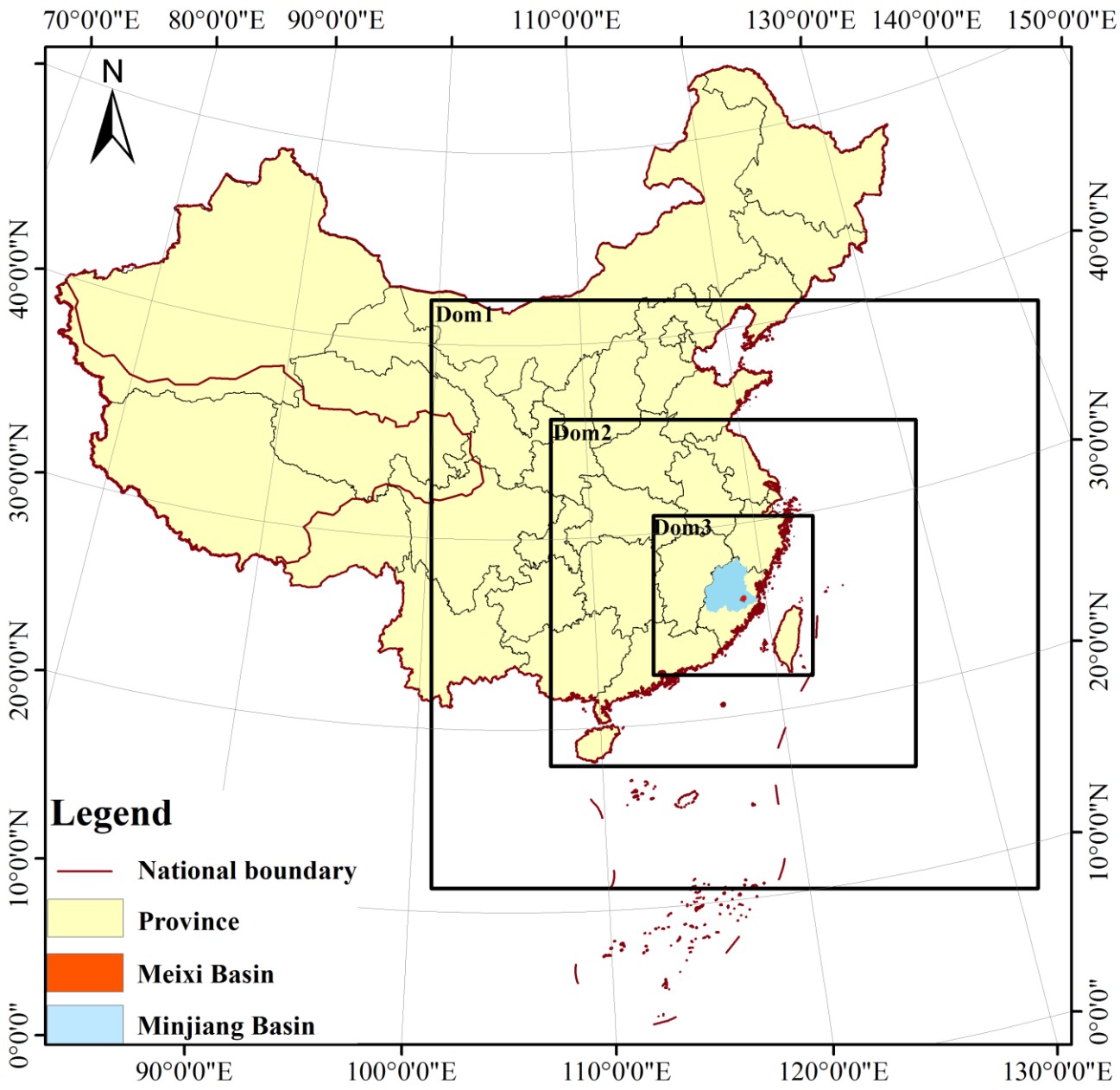

**Figure 1.** The location of the Meixi catchment and three nested domains.

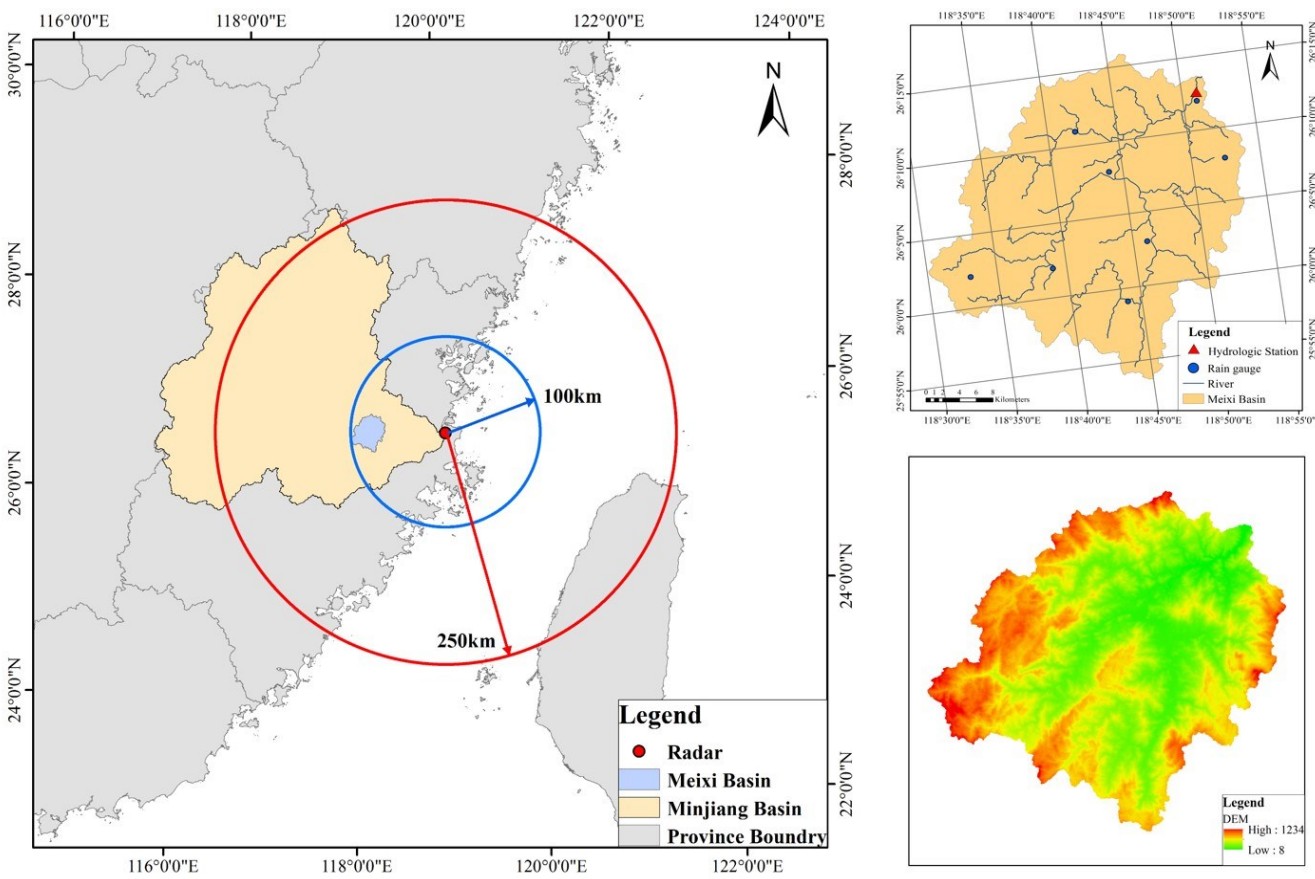

**Figure 2.** Radar scan area and Meixi basin.

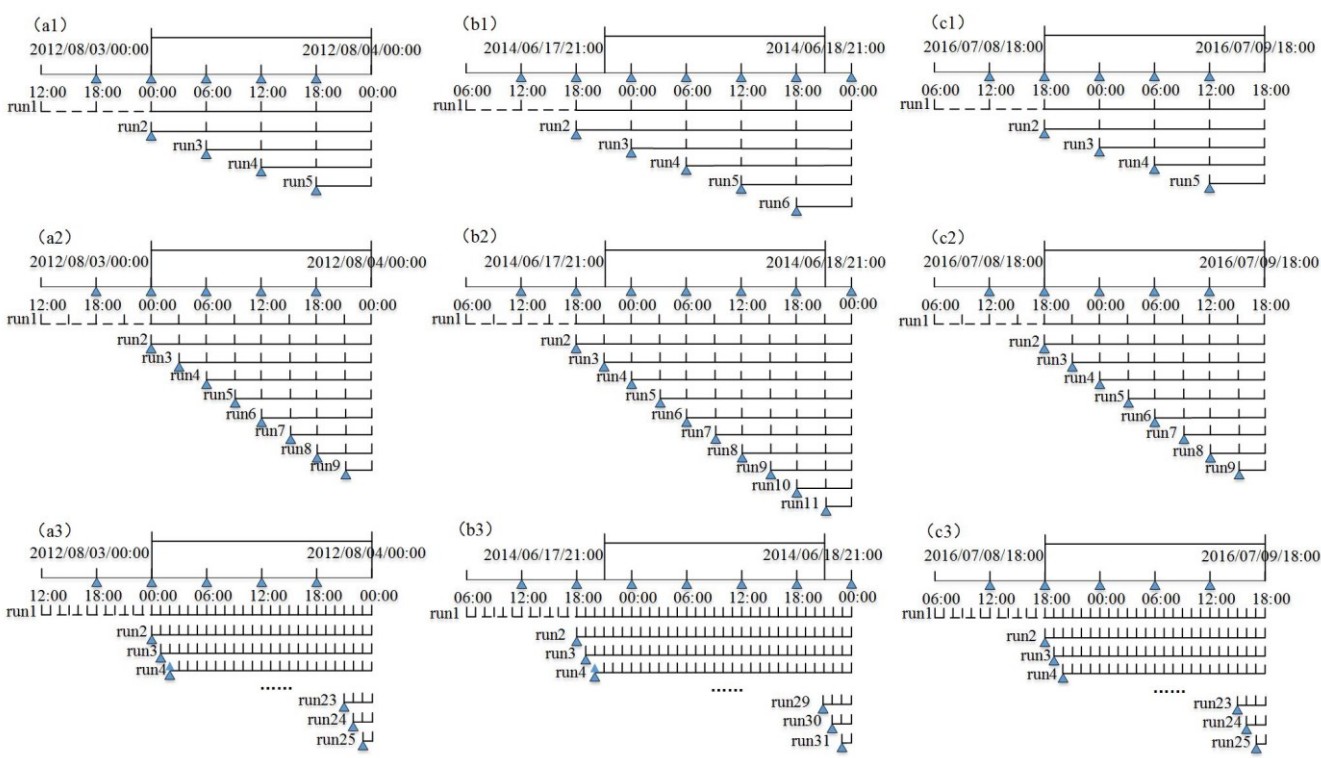

**Figure 3.** The time bars of the assimilation cycling runs for (a) storm event I, (b) storm event II and (c) storm event III.

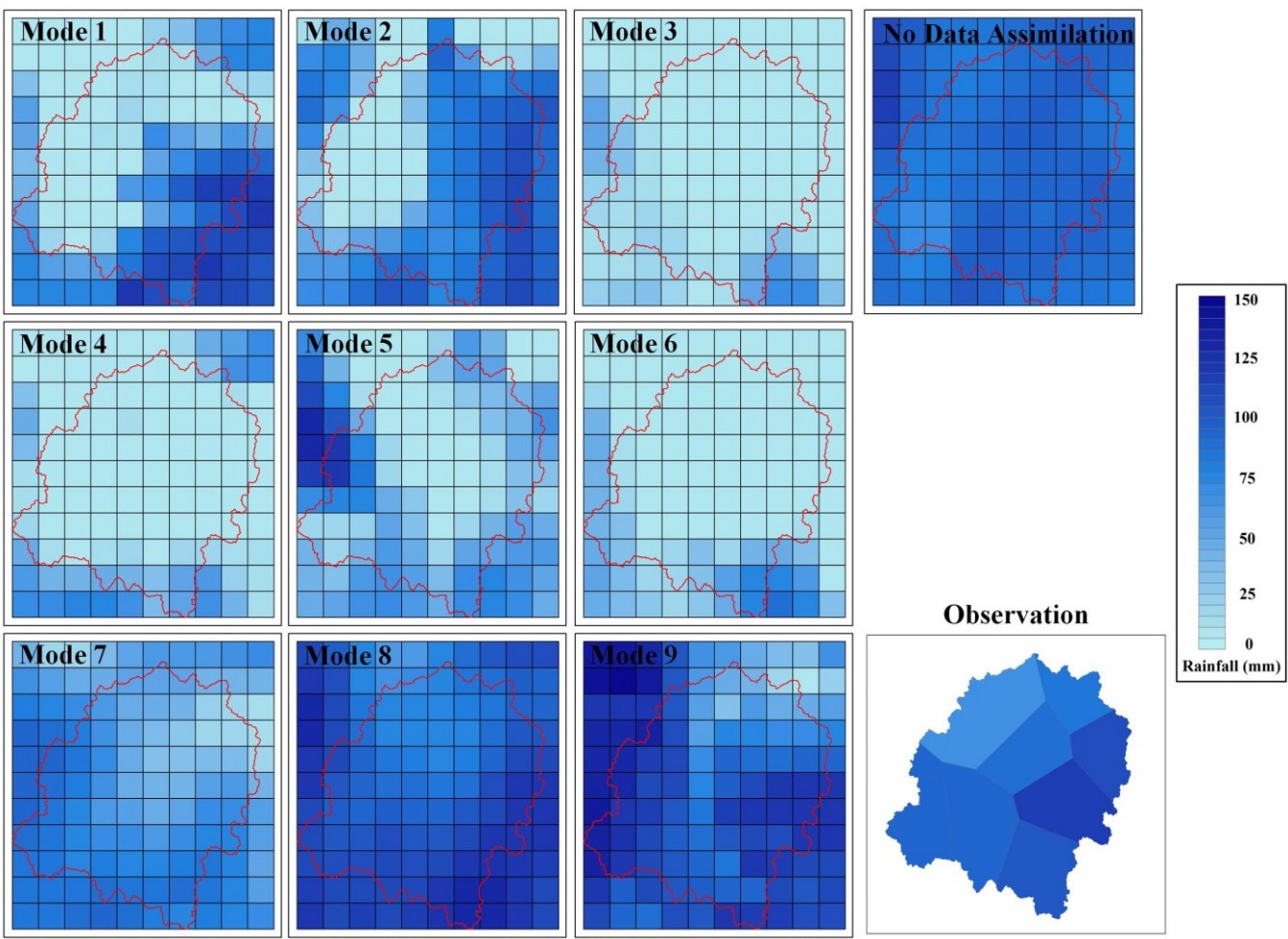

**Figure 4.** Spatial distribution of the simulated 24 h rainfall accumulations with nine data assimilation modes for Event I.

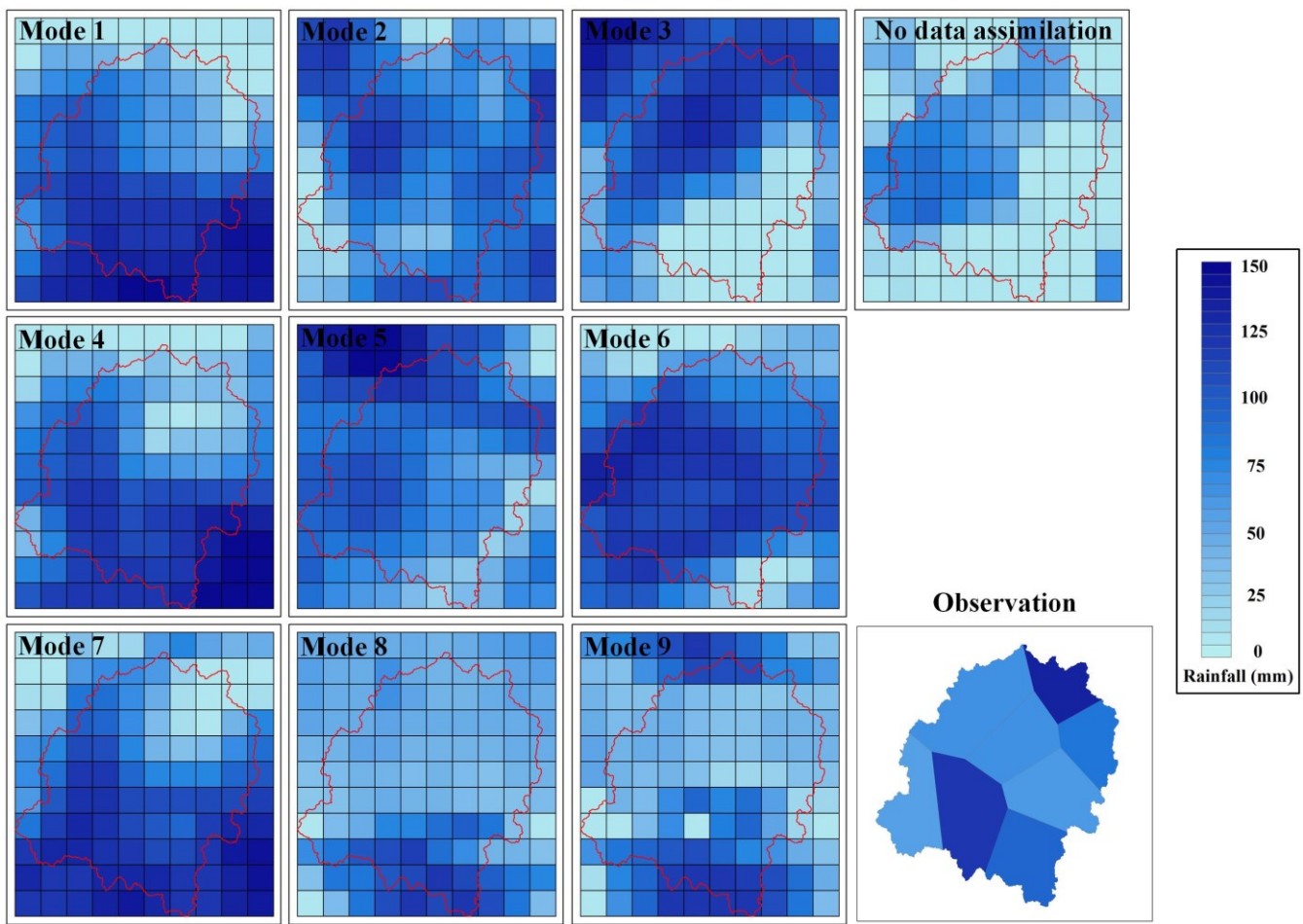

**Figure 5.** Spatial distribution of the simulated 24 h rainfall accumulations with nine data assimilation modes for Event II.

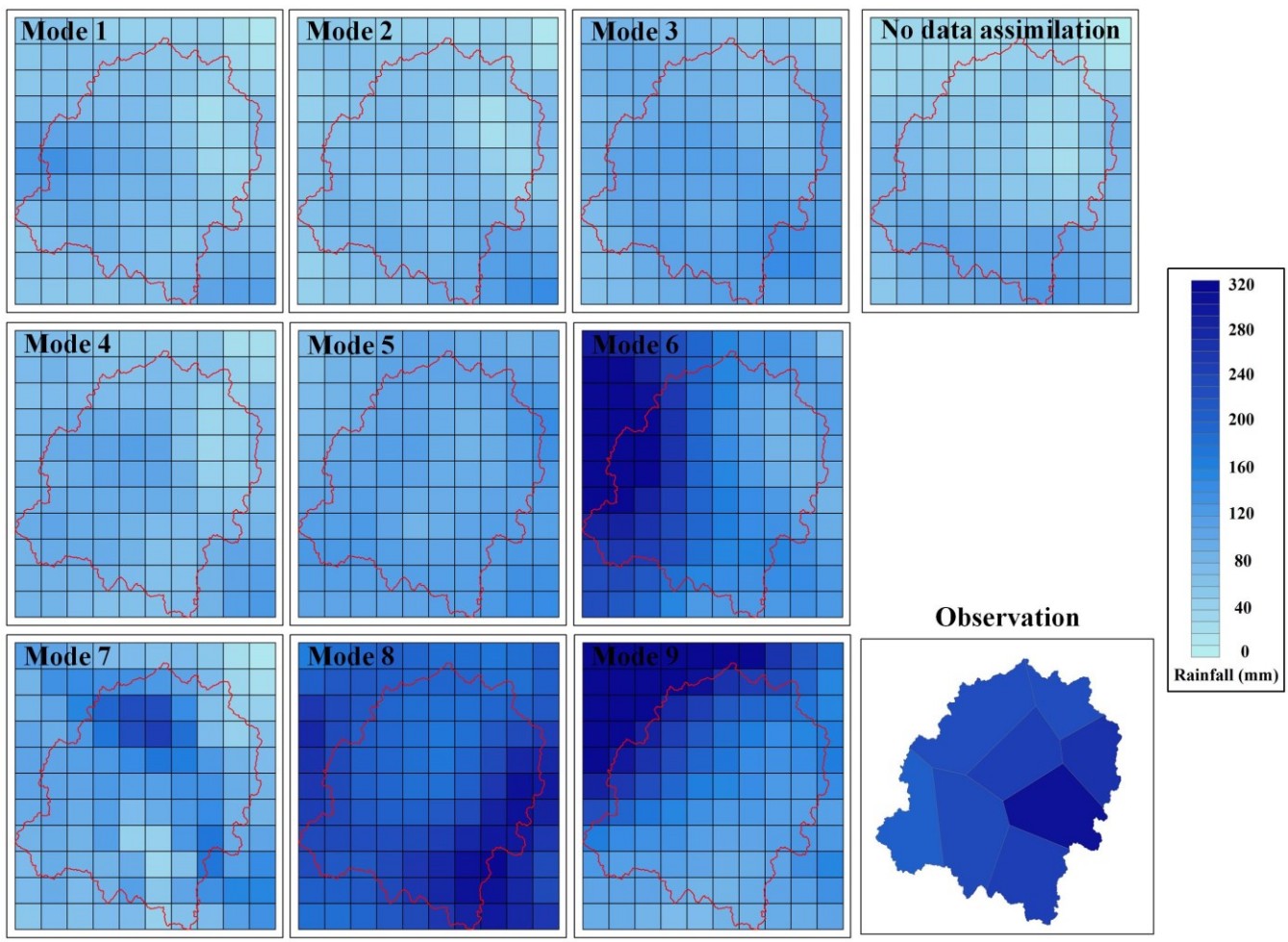

**Figure 6.** Spatial distribution of the simulated 24 h rainfall accumulations with nine data assimilation modes for Event III.

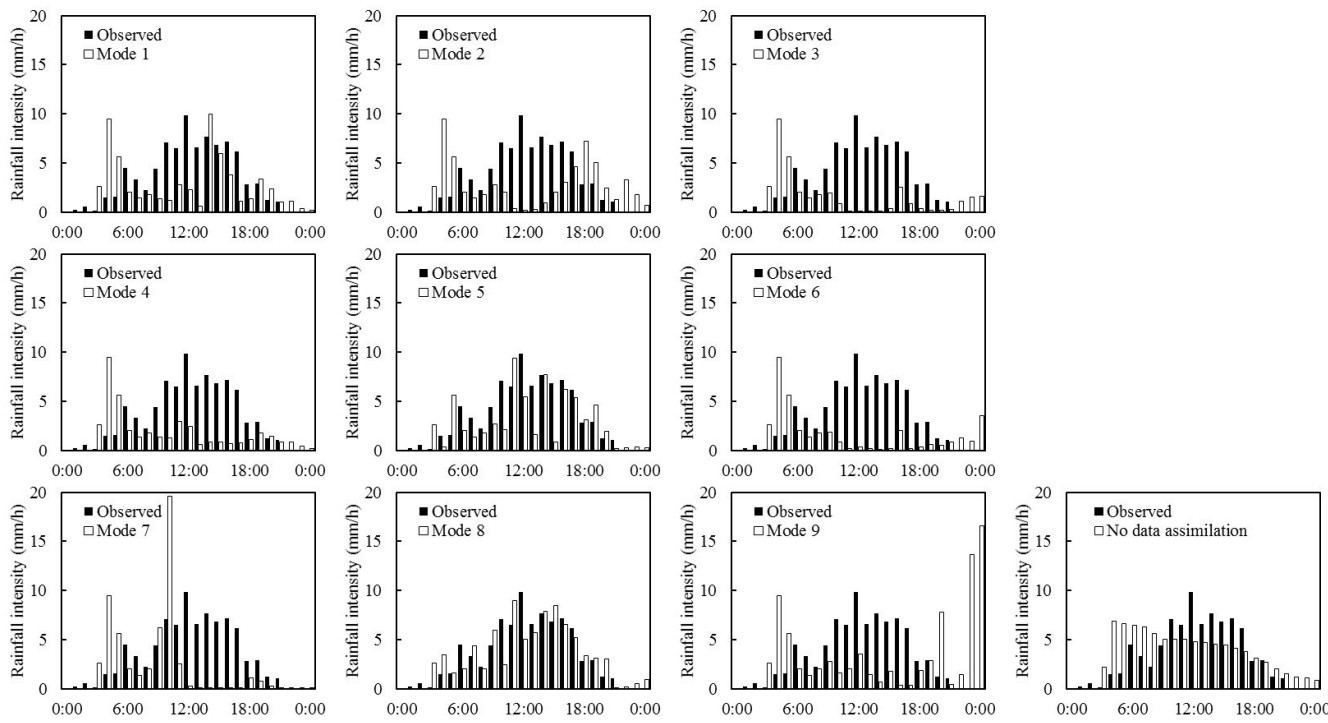

**Figure 7.** Time series bars of observed and simulated areal rainfall with nine data assimilation modes and the rainfall observation for Event I.

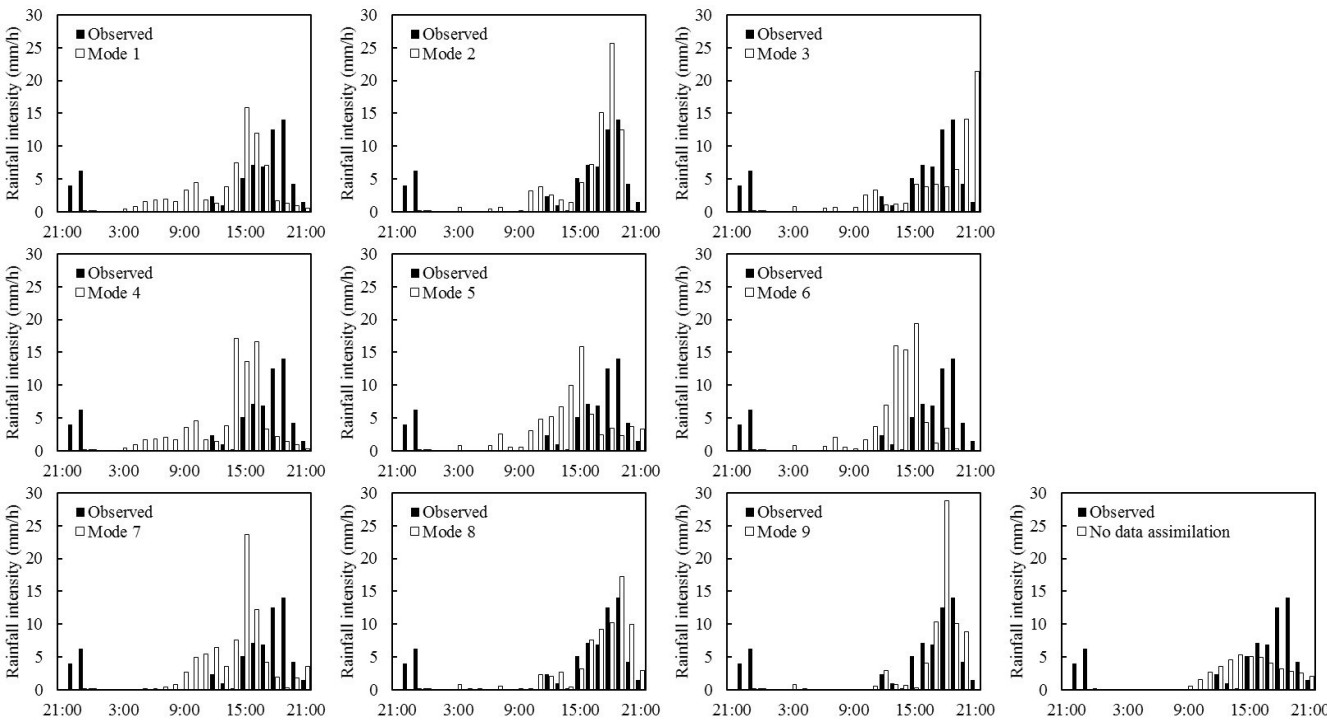

**Figure 8.** Time series bars of observed and simulated areal rainfall with nine data assimilation modes and the rainfall observation for Event II.

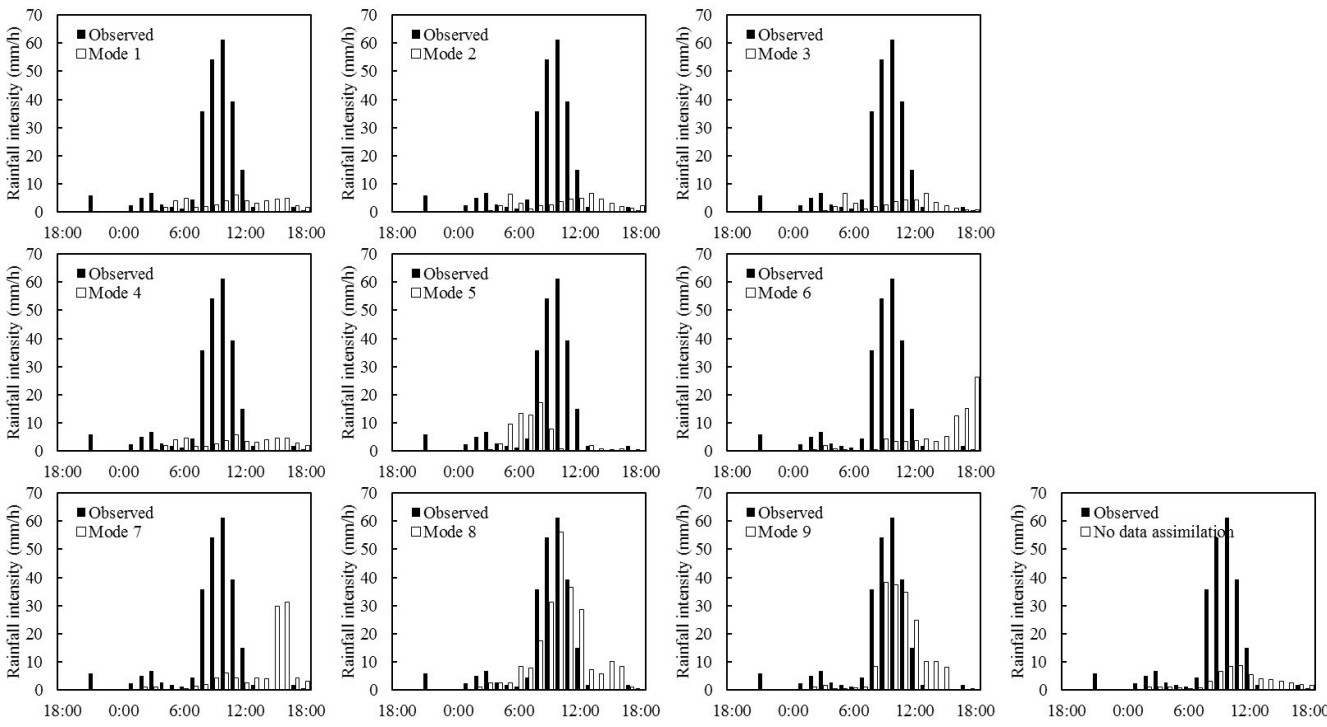

**Figure 9.** Time series bars of observed and simulated areal rainfall with nine data assimilation modes and the rainfall observation for Event III.

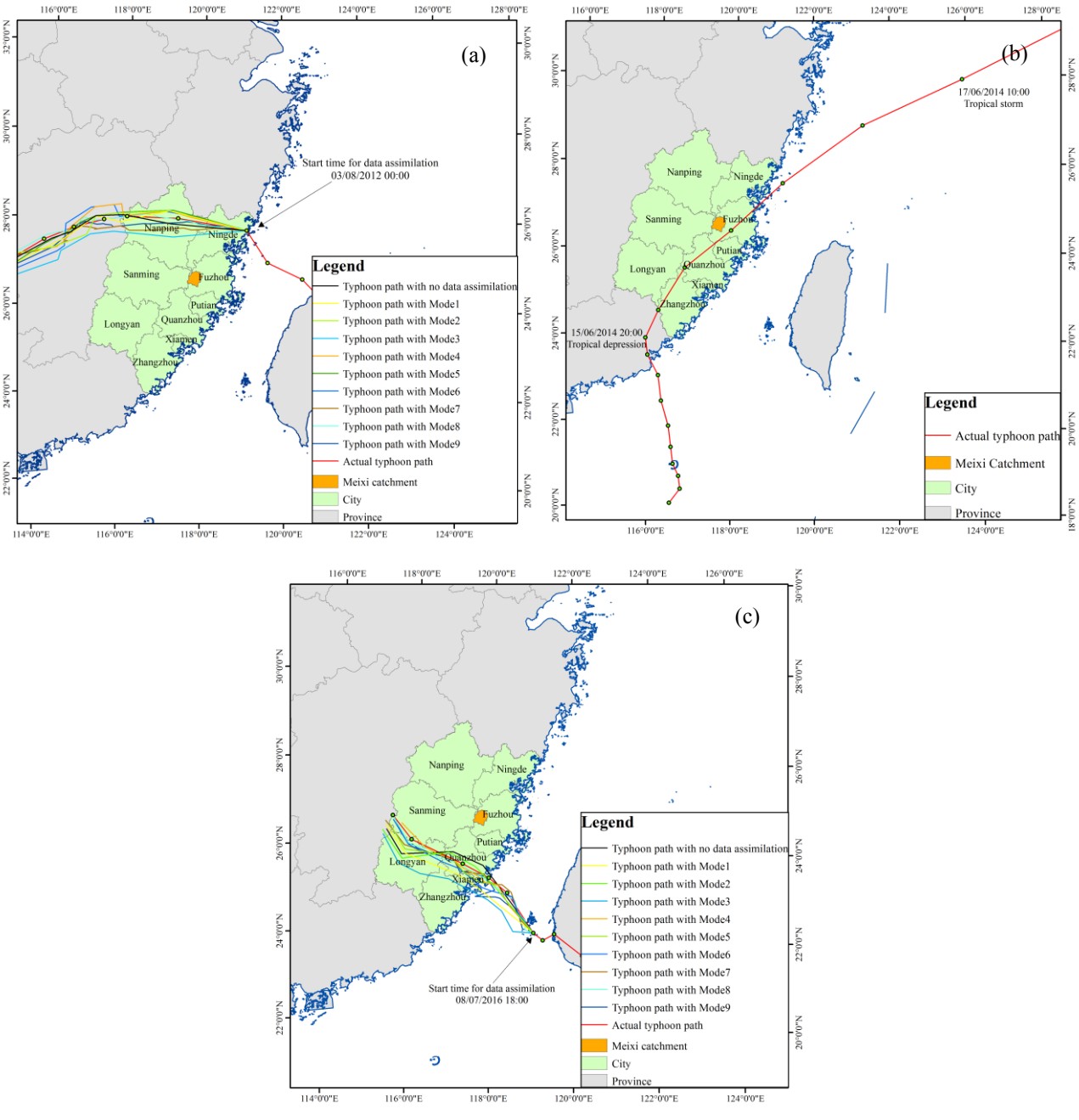

**Figure 10.** Typhoon path and simulations for (a) Saola, (b) Hagibis and (c) Nepartak.

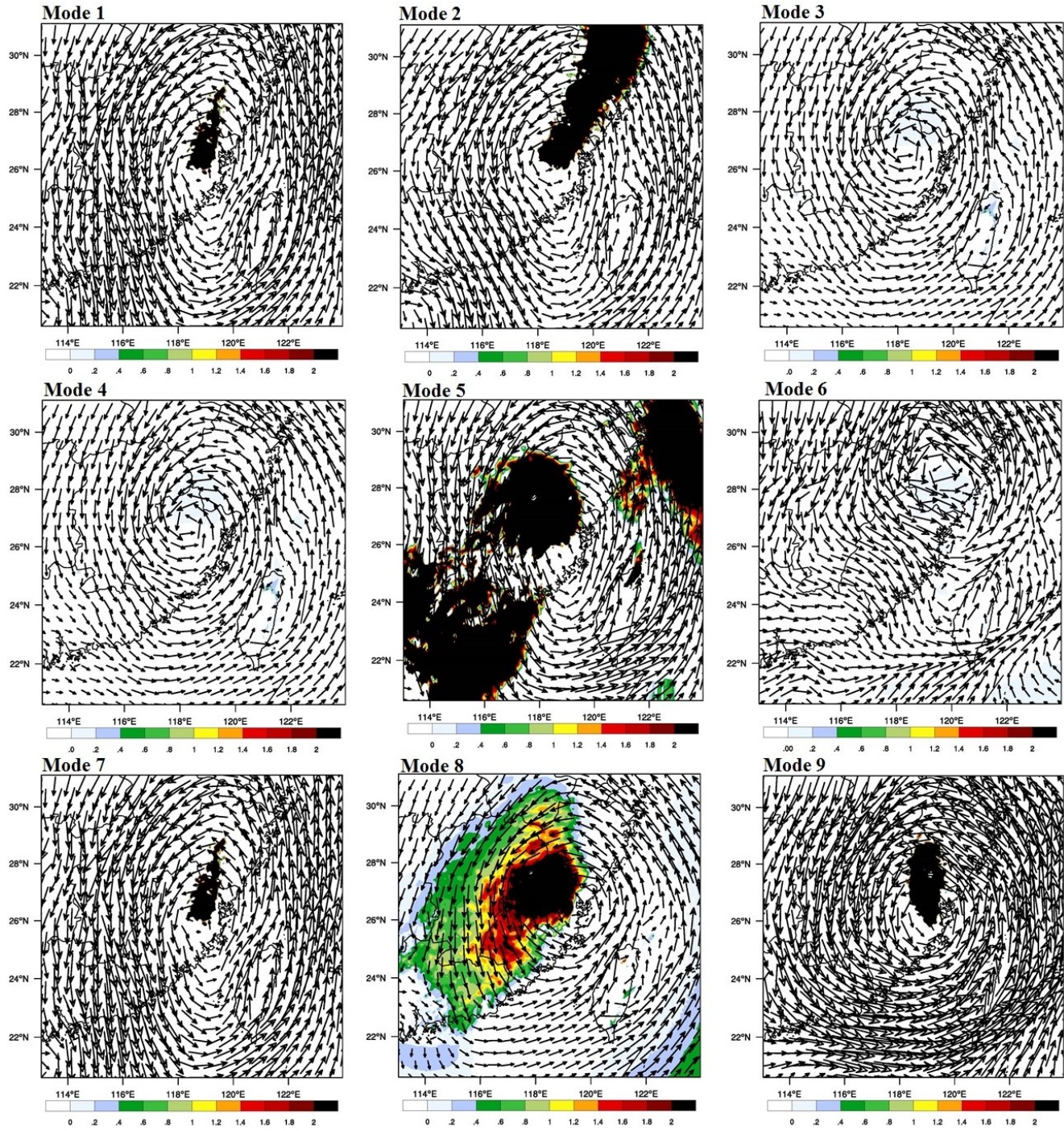

**Figure 11.** Wind field and water vapor transportation increment (850hPa) for Event I at 12:00 on August 3, 2012.

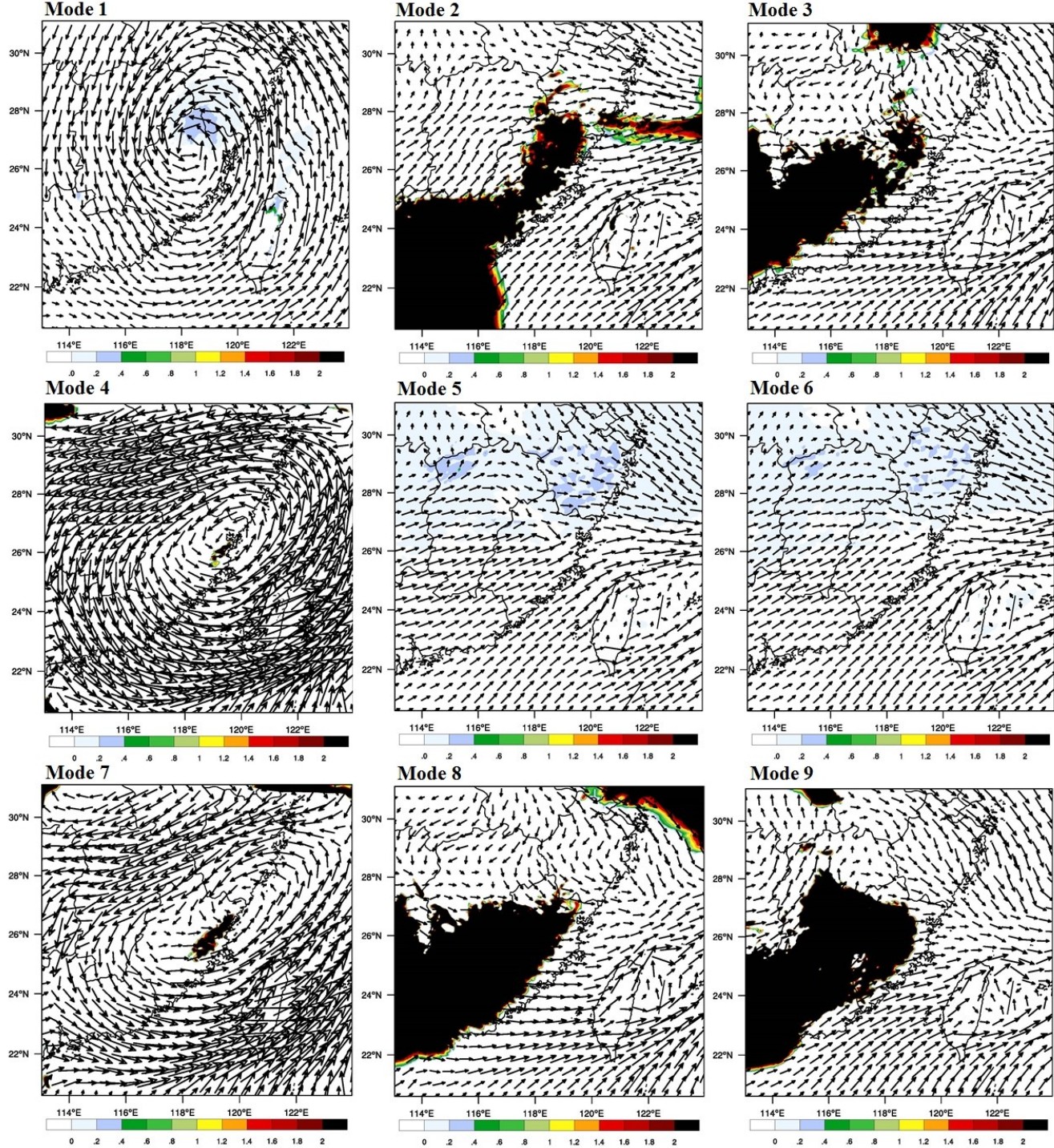

**Figure 12.** Wind field and water vapor transportation increment (850hPa) for Event II at 18:00 on June 18, 2014.

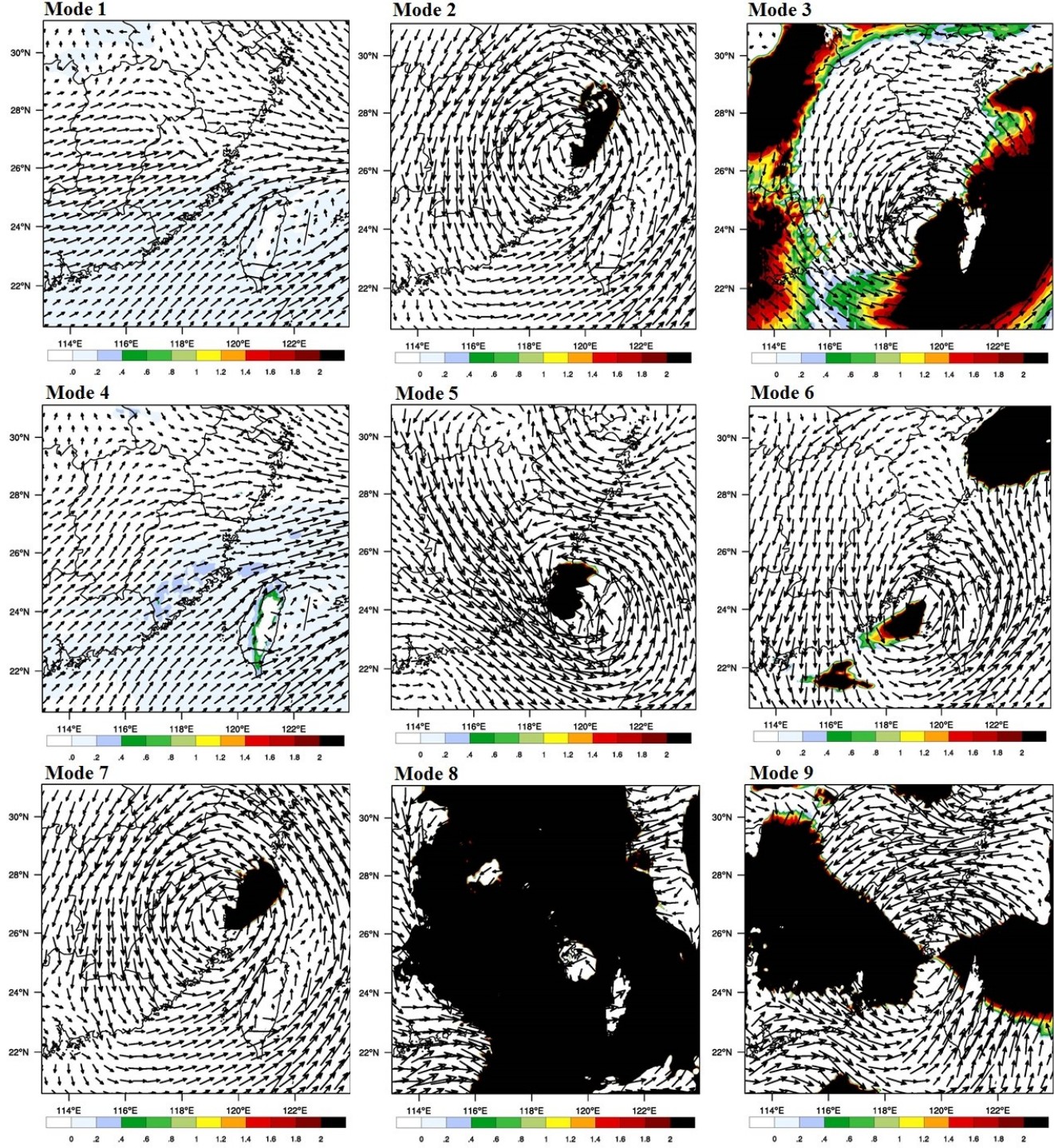

**Figure 13.** Wind field and water vapor transportation increment (850hPa) for Event III at 6:00 on July 9, 2016.