# Peer review of "Typhoon rainstorm simulations with radar data assimilation in southeast coast of China"

_Natural Hazards and Earth System Sciences, 2020_

## Referee Comment (RC1) · Anonymous Referee #1 · 21 Jul 2020

This study aims to improve the initial and lateral boundary conditions of the Numerical Weather Prediction (NWP) systems by assimilating Weather radar observations. Nine assimilation modes are designed by three kinds of radar data and three assimilation time intervals. The goal is noble to find the best option of the data assimilation mode for typhoon rainstorm simulations. The paper is well structured and understandable. However, the novelty of the work is still need to be highlighted. English usage in current manuscript should be improved for good readability. Further improvements and clarifications are needed before the paper is acceptable. Detailed comments are listed below:

Major comments: 1. Line 11, Page 5: Eq. (8), the RMSE is expressed as percentage? The numerator part is the RMSE and I think the equation is wrong. Please check as

well as Table * and *.

2. Why do you use FNL to drive the WRF model? Would using data from other centers like ECMWF change your final conclusion? As analysis data, FNL has also assimilated data, why do you not use GFS?

3. The rainfall is influenced by typhoon storms. Comparing the realistic typhoon path with the simulations can help to prove the accuracy of the assimilation results. Please add description and figures for the typhoon path simulations.

4. The results are encouraging that shortening the assimilation time interval can improve the rainfall simulations in most cases. How about half an hour or just 6 minutes? I suggest the authors do more work in further study. The research prospects can added in section 5 Discussion.

Minor comments:

1. Abstract, the 'radial velocity' is repetition

2. Plots showing the orography and the location of rain gauges would be desirable. You can add the information in Fig.2.

3. It would be helpful to summarize all physical parameterizations in a table.

4. Please provide references for the two evaluation statistics, CSI and RMSE.

5. Line 24, Page 5: km2 should be km2. Please correct.

6. Line 16, Page 10: add references for these data assimilation model.

---

## Referee Comment (RC2) · Anonymous Referee #2 · 27 Jul 2020

Review of "Typhoon rainstorm simulation with radar data assimilation in southeast coast of China" by Tian J., Liu R., Ding L., Guo L. and Zhang B. (Manuscript ID: NHESS-2020-146)

**Overall**:

This study exploits radar data assimilation into the Weather Research and Forecasting (WRF) model in order to improve the model's performance during typhoon rainfall simulations over a medium-scale catchment in southeast China. Nine different assimilation modes are examined based on the data and time intervals applied. Many previous studies have already examined the impact of assimilating different kind of radar data (i.e., radar reflectivity, radial velocity, radar reflectivity+radial velocity, as in this case) into NWP systems (e.g., Liu et al., 2018; Sugimoto et al., 2009). The novelty of the present study seems to lies on the investigation of three different assimilation time intervals (1h, 3h and 6h). However, this novelty is not highlighted sufficiently in the manuscript. Also, the paper lacks a concrete structure and an adequate methodological framework, while there are quite some occasions where the English could be improved. Detailed comments on the manuscript are provided below.

**Major**:

*Novelty*: Please elaborate more to demonstrate the novelty of the current study. Why the assimilation time intervals are important? How they affect the performance of the data assimilation? Why previous studies (for instance those you refer in the introduction; Page 2, L. 28-32 and Page 3, L. 1-2) follow different approaches (i.e., 1h, 3h, 6h)? How these time intervals have been set so far in the literature (more referenced are needed)? Based on experience? Is this the first study examining different assimilation time intervals?

*Structure*: The structure in Sections 1-3 is confusing for the reader. Firstly, in several parts of Sections 2-3 (e.g., Page 5, L. 12-14 and Page 6, L. 3-4), the motivation of conducting the study is mentioned. However, a comprehensive description of the background of the study (including the choice of the study area) should be given in Section 1 (Introduction). Secondly, the various information are mixed, as the model description (sub-sections 2.1, 2.2, 2.3) is presented with the evaluation process (sub-section 2.4) and then, the study area and storm events (sub-section 3.1), and numerical experiments (sub-sections 3.2. and 3.3) are presented. I strongly suggest, revising the above structure following a more appropriate set-up (for example: study area and case studies -> model description and numerical experiments -> evaluation process).

*Methodology*:

a) The authors highlight the need of accurate rainfall forecasts in the study area. Thus, I would expect examining the radar data assimilation options under an operational forecasting model configuration. However, they use the global analysis FNL data, which are not maintained in real-time, to drive the model instead of an operational real-time global dataset (e.g., NCEP GFS). Also concerning the model set-up, what do you mean by "considering the application effect and frequency in southeast coast of China?" (Page 3. L. 29-30)? How does it affect the selection of physics options? Please provide a more clear and sufficient background for justifying the applied model configuration.

b) Please justify the use of the Control Variable option 3 (CV3) of the WRF-3Dvar system for the model background errors covariance matrix (B matrix). As the authors acknowledge (e.g., Page 4, L. 10-11 and Page 10, L. 10-13), the B matrix has a strong impact on data assimilation process. Using domain-specific model background errors (i.e., CV5 option), instead of global (i.e., CV3 option), could lead to different results and conclusions. Since CV5 is a more appropriate option compared to CV3 option, and it is a common practice in data assimilation literature (e.g., radar data: Mazarella et al., 2019, conventional observations: Yang et al., 2014, and satellite and GNSS data: Giannaros et al., 2020; Lagasio et al., 2019), I strongly suggest conducting the study using the CV5 B matrix option.

c) The description of the evaluation process is unclear and insufficient. No information (map illustration, data temporal analysis and coverage etc.) is presented concerning the rain gauges (Page 5, L. 6-7) used for evaluating the model results. No information is presented concerning the method for pairing the model output and observations (e.g., nearest neighbor?). What do you mean by "areal rainfall observation at each rain gauge i" (Page 5, L. 15), since, the areal rainfall is calculated at the catchment scale using the observations from all 8 stations (Page 5, L. 6-7)? In overall, the terms "spatial" and "temporal" for computing the statistics CSI and RMSE are confusing. For example, to my understanding, spatial RMSE refers to the evaluation of the modeled 24-h rainfall considering all 8 stations, while temporal RMSE refers the evaluation of the basin-averaged rainfall using 24 model-observations pairs. However, both metrics consider the spatial dimension. Most studies in the literature apply the standard approach of domain-wide statistics (spatial dimension), using model-observation pairs of the examined variable (e.g., 1h or 24h rainfall) over all available stations, aggregated for certain time periods (temporal dimension). Thus, please consider revising the application of the statistics. Also, please consider computing more statistic metrics (e.g., POD, FAR etc.) to enhance the evaluation process. In the same direction, please consider evaluating the model results under different time intervals (e.g., 6h; 0, 6, 12, 18) and rain thresholds (e.g., >0.1, > 0.2 etc.). Finally, please provide information in the description of evaluation process concerning the construction and usage of Figures 4-9. For instance, do Figures 4-6 refer to the 24-h modeled and observed rainfall? Do Figures 7-9 refer to areal rainfall?

d) 2/3 typhoon events affected the study area as tropical cyclones and had a limited impact in the study area. This fact does not support the aim of the study, which focus of typhoon rainfall simulations. I suggest including more high-impact typhoon rainfall events in the study. Also, please refer in more detail to the impacts on properties, people etc. in the study catchment, as well as to the flooding mechanisms (fluvial?) in the area. This will assist the results interpretation in terms of natural hazard analysis.

*Results/Discussion*: Please provide evidence on how assimilating radial velocity and radar reflectivity affect the WRF model's initial and boundary conditions (ICBC), and performance during the conducted numerical experiments. For example, you could compare the ICBC wind field and water vapor transportation between the experiments. This is important to support the interpretation of the results.

**Minor:**

Page 2, L. 22-27, 29-31 and Page 2, L. 1-2: Please refer to models and data assimilation schemes used.

Page 4, Section 2.2.3: The description could be improved in terms of English and details provided.

Please refer to what is being shown in Figures 4-9 (24-h rainfall? Areal rainfall? See comment c) in Methodology above).

Please enhance the resolution of Figures 7-9.

**Below there are some examples where the English could be improved**.

Title: Please replace "simulation" by "simulations"

Page 2, L. 2 and 5: Please replace "system" by "systems"

Page 2, L. 10: Please replace "by" by "using the"

Page 2, L. 14: Please replace "WRF-LTNGA" by "the WRF-LTNGA scheme"

Page 2, L. 16-17: Please move "for hydrological applications" to the previous sentence ("...into the WRF model for hydrological applications")

Page 2, L. 19-20: Please rephrase

Page 2, L. 21: Please replace "the" by "their"

Page 2, L. 28: Please remove "the"

Page 3, L. 5: Please change to "caused by the interaction of typhoons and the complex terrain"

Page 3, L. 6-7: Please rephrase.

Page 3, L. 11: Please rephrase "flood disasters have attacked..."

Page 3, L. 23: Please replace "can be" by "is"

Page 3, L. 28: Please add "a" ("...has a significant effect...")

Page 5, L. 13-14: Please rephrase "... and 24 for N is the ..."

Page 6, L. 1-2: Please use past tense.

etc.

References

Giannaros, C., Kotroni, V., Lagouvardos, K., Giannaros, M.T., Pikridas, C., 2020. Assessing the Impact of GNSS ZTD Data Assimilation into the WRF Modeling System during High-Impact Rainfall Events over Greece. Remote Sens. . https://doi.org/10.3390/rs12030383

Lagasio, M., Pulvirenti, L., Parodi, A., Boni, G., Pierdicca, N., Venuti, G., Realini, E., Tagliaferro, G., Barindelli, S., Rommen, B., 2019. Effect of the ingestion in the WRF model of different Sentinel-derived and GNSS-derived products: analysis of the forecasts of a high impact weather event. Eur. J. Remote Sens. 52, 16–33. https://doi.org/10.1080/22797254.2019.1642799

Liu, J., Tian, J., Yan, D., Li, C., Yu, F., Shen, F., 2018. Evaluation of Doppler radar and GTS data assimilation for NWP rainfall prediction of an extreme summer storm in northern China: From the hydrological perspective. Hydrol. Earth Syst. Sci. 22, 4329–4348. https://doi.org/10.5194/hess-22-4329-2018

Mazzarella, V., Maiello, I., Ferretti, R., Capozzi, V., Picciotti, E., Alberoni, P.P., Marzano, F.S., Budillon, G., 2020. Reflectivity and velocity radar data assimilation for two flash flood events in central Italy: A comparison between 3D and 4D variational methods. Q. J. R. Meteorol. Soc. 146, 348–366. https://doi.org/10.1002/qj.3679

Sugimoto, S., Andrew Crook, N., Sun, J., Xiao, Q., Barker, D.M., 2009. An examination of WRF 3DVAR radar data assimilation on its capability in retrieving unobserved variables and forecasting precipitation through observing system simulation experiments. Mon. Weather Rev. 137, 4011–4029. https://doi.org/10.1175/2009MWR2839.1

Yang, J., Duan, K., Wu, J., Qin, X., Shi, P., Liu, H., Xie, X., Zhang, X., Sun, J., 2015. Effect of data assimilation using WRF-3DVAR for heavy rain prediction on the northeastern edge of the Tibetan Plateau. Adv. Meteorol. 2015. https://doi.org/10.1155/2015/294589

---

## Author Comment (AC1) · 18 Nov 2020

Point 1: Line 11, Page 5: Eq. (8), the RMSE is expressed as percentage? The numerator part is the RMSE and I think the equation is wrong. Please check as well as Table * and *.

Reply: Thanks for the reviewer's suggestion. The values in Table 6 and 7 is right. Equation 8 has been revised. The sentences in Line 4-6, Page 7 are revised as: "The spatiotemporal patterns of the rainfall simulation are evaluated by the critical success index (CSI) and modified root mean square error (m-RMSE), which is defined as the ratio of root mean square error (RMSE) to the mean values of the corresponding observations (Prakash et al., 2014; Agnihotri and Dimri, 2015)"

[Figure]

Point 2: Why do you use FNL to drive the WRF model? Would using data from other centers like ECMWF change your final conclusion? As analysis data, FNL has also assimilated data, why do you not use GFS?

Reply: We appreciate the referee's deep insights. The initial and lateral boundary conditions provided by different centres like NCEP, ECMWF and CMA may make some difference of the rainfall forecasts. Some studies have specialised the different performances of the WRF model based on the initial and lateral boundary conditions from the different centres (Zhao et al, 2012; Zhang et al, 2018). Before the NCEP data was used in this study, we also tests ECMWF for data assimilation with storm events in the same region. Although the rainfall forecasts showed a little different, the patterns of improvements from different data assimilation modes were quite similar and the same conclusions can be obtained. In order to highlight the main purpose of this study, we only present the assimilation results using the FNL. We hope our work can inspire further studies on testing the data assimilation effects using other boundary data, such as ECMWF. For further clarification, though FNL assimilates meteorological data with low resolution, local observations such as radar data with high resolution are not included and FNL can hardly simulate the rainfall in meso-and small-scale systems. Many studies indicate radar data assimilation can improve the rainfall simulation significantly. FNL has higher applicability and accuracy than GFS for historical events simulation. GFS with no data assimilation is always used for weather forecasting. That is why we use FNL not GFS.

The following sentences are added in Line 34, Page 12 and Line 1-3, Page 13: "In reality, ECMWF is also tested for the data assimilation before FNL is used in this study. Although the rainfall simulations show some differences based on the two kinds of boundary conditions, the patterns of improvements from different data assimilation modes are quite similar and the same conclusions can be obtained." References: Zhao P. K., Wang B., Liu J., et al. A DRP–4DVar data assimilation scheme for typhoon initialization using sea level pressure data, Mon. Weather Rev., 140, 1191-

1203, doi: 10.1175/MWR-D-10-05030.1, 2012. Zhang X., Xiong Z., Zheng J., et al. High-resolution precipitation data derived from dynamical downscaling using the WRF model for the Heihe River Basin, northwest China. Theor. Appl. Climatol., 131, 1249-1259, doi: 10.1007/s00704-017-2052-6, 2018.

Point 3: The rainfall is influenced by typhoon storms. Comparing the realistic typhoon path with the simulations can help to prove the accuracy of the assimilation results. Please add description and figures for the typhoon path simulations.

Reply: The description and figures for typhoon path simulations are added in the manuscript. According to the simulations of Saola and Nepartak, the accurate typhoon path simulation always leads to accurate rainfall simulation. However, for typhoon Hagibis, when WRF model assimilates the radar data, the actual typhoon center is far away from the Meixi catchment. Hence, the typhoon path simulations cannot help to prove the accuracy of the rainfall simulations for different assimilation modes. The actual typhoon path for Hagibis is added. In addition, the wind field and water vapor transportation for different modes are also compared in the manuscript to support the interpretation of the results.

The following sentences are added in Line 2-5, Page 11: "In order to prove the accuracy of the assimilation results, typhoon paths for different assimilation modes are also simulated (Fig. 10). According to the simulations of Saola and Nepartak, the accurate typhoon path simulation always leads to accurate rainfall simulation. However, for typhoon Hagibis, when WRF model assimilates the radar data, the actual typhoon center is far away from the Meixi catchment. Hence, only the actual typhoon path for Hagibis is added."

Point 4: The results are encouraging that shortening the assimilation time interval can improve the rainfall simulations in most cases. How about half an hour or just 6 minutes? I suggest the authors do more work in further study. The research prospects can added in section 5 Discussion.

Reply: Thanks for the reviewer's suggestion. According to the manuscript, assimilating radial velocity with time interval of 1 h can significantly improve the rainfall simulations and the REs are all lower than 10%. The rainfall simulations are satisfactory for flood forecasting in small and medium basins. That is why we have no further reduction in the assimilation time interval. However, as the reviewer mentioned, shortening the assimilation time interval can improve the rainfall simulations and further shortening the assimilation interval is worth exploring. On the one hand, radial velocity can correct the initial and lateral boundary condition more timely with higher assimilation frequency, and the rainfall simulations should be better in terms of theory. On the other hand, the observation errors of radial velocity may be amplified with high assimilation frequency in WRF model. There may be an "inverted u" relationship between accuracy of rainfall simulation and assimilation time interval (Myung et al., 2009). Further study should be carried out to investigate the optimal assimilation time interval.

The following sentences are added in Line 7-11, Page 12: "Do further shortening the assimilation interval obtain better rainfall simulation? In terms of theory, the answer is yes, because improving the assimilation frequency can correct the initial and lateral boundary condition timely. However, the observation errors of radial velocity may be amplified with high assimilation frequency in WRF model. There may be an "inverted u" relationship between accuracy of rainfall simulation and assimilation time interval (Myung et al., 2009). Further study should be carried out to investigate the optimal assimilation time interval." References: Myung, J. I. The importance of complexity in model selection, J. Math. Psychol., 44, 190-204, doi:10.1006/jmps.1999.1283, 2000.

Point 5: Abstract, the 'radial velocity' is repetition.

Reply: Revised. The repetitions are removed.

Point 6: Plots showing the orography and the location of rain gauges would be desirable. You can add the information in Fig.2.

Reply: As the reviewer mentioned, the orography and the location of rain gauges are

added in Fig.2.

Point 7: It would be helpful to summarize all physical parameterizations in a table.

Reply: The physical parameterizations are listed in table 1. Physical parameterization Scheme Microphysics WRF Single-Moment 6 (WSM 6) Planetary boundary layer (PBL) Yonsei University (YSU) Longwave and shortwave radiation Rapid Radiative Transfer Model for application to GCMs (RRTMG) Land-surface model (LSM) Noah Cumulus Kain-Fritsch (KF)

Point 8: Please provide references for the two evaluation statistics, CSI and RMSE.

Reply: Two references are added. References: Agnihotri G., Dimri A. P. Simulation study of heavy rainfall episodes over the southern Indian peninsula, Meteorol. Appl., 22, 223-235, doi: 10.1002/met.1446, 2015. Prakash S., Sathiyamoorthy V., Mahesh C., et al. An evaluation of high-resolution multisatellite rainfall products over the Indian monsoon region, Int. J. Remote Sens., 35, 3018-3035, doi: 10.1080/01431161.2014.894661, 2014.

Point 9: Line 24, Page 5: km2 should be km2. Please correct.

Reply: Revised.

Point 10: Line 16, Page 10: add references for these data assimilation model.

Reply: Two references are added. References: Shen F., Min J., Xu D. Assimilation of radar radial velocity data with the WRF Hybrid ETKF-3DVAR system for the prediction of Hurricane Ike (2008), Atmos. Res., 169, 127-138, doi: 10.1016/j.atmosres.2015.09.019, 2016. Wang X., Barker D. M., Snyder C., et al. A hybrid ETKF–3DVAR data assimilation scheme for the WRF model. Part II: real observation experiments, Mon. Weather Rev., 136, 5132-5147, doi: 10.1175/2008MWR2445.1, 2012.

2020-146, 2020.
Interactive
comment

$$m\text{-}RMSE = \frac{\sqrt{\frac{1}{M}\sum_{j=1}^{M}\left(P_j' - P_j\right)^2}}{\frac{1}{M}\sum_{j=1}^{M}P_j}$$

**Fig. 1.** Equation 8

[Figure]

Figure 10. Typhoon path and simulations for (a) Saola, (b) Hagibis and (c) Nepartak.

**Fig. 2.** Figure 10

[Figure]

**Figure 2.** Radar scan area and Meixi basin.

**Fig. 3.** Figure 2

**Table 1.** Physical parameterizations used in the WRF model.

| Physical parameterization | Scheme |
|---|---|
| Microphysics | WRF Single-Moment 6 (WSM 6) |
| Planetary boundary layer (PBL) | Yonsei University (YSU) |
| Longwave and shortwave radiation | Rapid Radiative Transfer Model for application to GCMs (RRTMG) |
| Land-surface model (LSM) | Noah |
| Cumulus | Kain-Fritsch (KF) |

**Fig. 4.** Table 1

---

## Author Comment (AC2) · 18 Nov 2020

We appreciate very much the referee's insightful comments and helpful suggestions for our manuscript. Efforts have been made to address every point of the referee's concerns. Gramma mistakes and spelling errors are carefully be checked before the revision is finally submitted. With the help of the referee, we hope the revised manuscript can be found rigorously and sufficiently improved.

Major comments:

Point 1: Please elaborate more to demonstrate the novelty of the current study. Why the assimilation time intervals are important? How they affect the performance of the data assimilation? Why previous studies (for instance those you refer in the introduction;

Page 2, L. 28-32 and Page 3, L. 1-2) follow different approaches (i.e., 1h, 3h, 6h)? How these time intervals have been set so far in the literature (more referenced are needed)? Based on experience? Is this the first study examining different assimilation time intervals?

Reply: The suggestion is very important for demonstrating the novelty of the study. For the rainfall forecasting in catchment scale, the assimilation time intervals are important. The operational forecast from meteorological department is guidance forecast with a large forecasting area. It is impossible to focus on the accuracy of the rainfall in small and medium catchment scale. Limited computing power makes that the number of restarting the forecasting system is only 2-4 times per day. The forecasting accuracy descends gradually as the run time goes on, because the data assimilation is not in real time. Due to the poor accuracy in small scale and low-resolution, the rainfall forecasting from the meteorological department cannot be used directly as the input for hydrological forecasting in small and medium catchment. The local meteorological observations are necessary to be assimilated to improve the high resolution rainfall forecast. The NWP model maybe not corrected timely with long time interval of data assimilation, while shortening the time interval need a lot of computational resources and the observation errors in local meteorological observations may be also amplified with high assimilation frequency in NWP model.

In previous studies, the time interval of data assimilation is set based on experience or computing resources. Most studies focus on the assimilated data selection and assimilation algorithm. Few studies pay attention on the time interval of data assimilation. That is why we design nine different modes to investigate the reasonable use of radar data assimilation. The following sentences are added in Line 30-32, Page 2:

"Most studies focus on the assimilation algorithm and data selection. However, consistent conclusions have not been obtained for the option of radar reflectivity and radial velocity, and few studies pay attention on the time interval setting of data assimilation."

The following sentences are added in Line 7-15, Page 3:

"In reality, the operational forecast from meteorological department is guidance forecast with a large forecasting area. It is impossible to focus on the accuracy of the rainfall in small and medium catchment scale. Limited computing power makes that the number of restarting the forecasting system is only 2-4 times per day (Xie et al., 2016). The forecasting accuracy descends gradually as the run time goes on, because the data assimilation is not in real time. Due to the poor accuracy in small scale and low-resolution, the rainfall forecasting from the meteorological department cannot be used directly as the input for hydrological forecasting in small and medium catchment (Tian et al., 2019). The local meteorological observations are necessary to be assimilated to improve the high resolution rainfall forecast. The NWP model maybe not corrected timely with long time interval of data assimilation, while shortening the time interval need a lot of computational resources and the observation errors in local meteorological observations may be also amplified with high assimilation frequency in NWP model."

References: Xie, Y., Xing, J., Shi, J., Dou, Y., Lei, Y. Impacts of radiance data assimilation on the Beijing 7.21 heavy rainfall, Atmos. Res., 169, 318-330, doi: 10.1016/j.atmosres.2015.10.016, 2016. Tian, J., Liu, J., Yan, D., Ding, L., Li, C. Ensemble flood forecasting based on a coupled atmospheric-hydrological modeling system with data assimilation, Atmos. Res., 224, 127-137, doi: 10.1016/j.atmosres.2019.03.029, 2019.

Point 2: The structure in Sections 1-3 is confusing for the reader. Firstly, in several parts of Sections 2-3 (e.g., Page 5, L. 12-14 and Page 6, L. 3-4), the motivation of conducting the study is mentioned. However, a comprehensive description of the background of the study (including the choice of the study area) should be given in Section 1 (Introduction). Secondly, the various information are mixed, as the model description (sub-sections 2.1, 2.2, 2.3) is presented with the evaluation process (sub-section 2.4) and then, the study area and storm events (sub-section 3.1), and numerical experiments (sub-sections 3.2. and 3.3) are presented. I strongly suggest, revising the above structure following a more appropriate set-up (for example: study area and case studies -> model description and numerical experiments -> evaluation process).

Reply: Thanks for the reviewer's suggestion. The background of the choice of the study area is added in Line 24-26, Page 3:

"On July 9, 2016, heavy rainfall caused by typhoon Nepartak leads to severe flood, and attracted strong interest from the public, academics and government. Accurate rainfall simulation has a great practical significance in the study area."

Other description of the background of the study can be found in the reply of Point 1. The structure of the manuscript is revised accordingly. The section 2 is study area and case studies. The section 3 is model description and numerical experiments. The section 4 is rainfall evaluation statistics.

Point 3: The authors highlight the need of accurate rainfall forecasts in the study area. Thus, I would expect examining the radar data assimilation options under an operational forecasting model configuration. However, they use the global analysis FNL data, which are not maintained in real-time, to drive the model instead of an operational real-time global dataset (e.g., NCEP GFS). Also concerning the model set-up, what do you mean by "considering the application effect and frequency in southeast coast of China?" (Page 3. L. 29-30)? How does it affect the selection of physics options? Please provide a more clear and sufficient background for justifying the applied model configuration.

Reply: We appreciate the referee's deep insights. Firstly, FNL has higher applicability and accuracy than GFS for historical events simulation. GFS with no data assimilation is always used for weather forecasting. The aim of this study is to explore the reasonable use of Doppler radar data assimilation to improve the rainfall simulation rather than rainfall forecast in real-time. That is why we use FNL not GFS. Secondly, the selection of physics is investigated in our previous study, which has been published recently

(Tian et al., 2020). Thirty-six physical parameterization combinations are designed by three microphysics, three pairs of longwave/shortwave radiations and four cumulus parameterizations. The physical parameterizations in best performance are used in this study. The sentences in Line 29-31, Page 4 and Line 1-2, Page 5 are revised as:

"Considering the application effect in southeast coast of China and also according to our previous research, WRF Single-Moment 6 (WSM 6) for microphysics, Yonsei University (YSU) for PBL, Rapid Radiative Transfer Model for application to GCMs (RRTMG) for longwave and shortwave radiation, Noah for LSM and Kain-Fritsch (KF) for cumulus physics are adopted in this study (Srivastava et al., 2015; Hazra et al., 2017; Cai et al., 2018; Tian et al., 2020)."

References: Tian, J., Liu, R., Ding, L., Guo, L., Liu, Q. Evaluation of the WRF physical parameterisations for Typhoon rainstorm simulation in southeast coast of China, Atmos. Res., 247, 105130, doi: 10.1016/j.atmosres.2020.105130, 2020.

Point 4: Please justify the use of the Control Variable option 3 (CV3) of the WRF-3Dvar system for the model background errors covariance matrix (B matrix). As the authors acknowledge (e.g., Page 4, L. 10-11 and Page 10, L. 10-13), the B matrix has a strong impact on data assimilation process. Using domain-specific model background errors (i.e., CV5 option), instead of global (i.e., CV3 option), could lead to different results and conclusions. Since CV5 is a more appropriate option compared to CV3 option, and it is a common practice in data assimilation literature (e.g., radar data: Mazarella et al., 2019, conventional observations: Yang et al., 2014, and satellite and GNSS data: Giannaros et al., 2020; Lagasio et al., 2019), I strongly suggest conducting the study using the CV5 B matrix option.

Reply: I agree with the reviewer's point that different B matrix has a strong impact on rainfall simulation. The choice of the B matrix even constructing the B matrix are worth to be investigated, and B matrix is the key field of data assimilation. However, there is still no unified conclusion to make clear that which B matrix is better or how to construct the B matrix can make the simulation more close to the reality. The main reason is that none of them can accommodate all synoptic situation in different regions. CV5 B matrix is also need to be tested. Liu et al. (2013) explores the effect of data assimilation by WRF-3DVar with CV3 for different types of rainfall simulation in catchment scale in southwest England, and the results show that data assimilation with CV3 can also obtain accurate rainfall simulation. Wang et al. (2013) develops an indirect radar reflectivity assimilation scheme within WRF 3D-Var to improve the heavy rainfall simulations in Beijing, and CV5 is used. The results show that the assimilation scheme improves the subsequent prediction of the location and intensity of rainfall. In this study, assimilating radial velocity with time interval of 1 h has been able to obtain satisfactory rainfall simulations for three different storm events in the case of using CV3. Some studies also indicate that the simulation trend of various numerical experimentations will not change with different B matrixes (Blni et al., 2015). The main purpose of this study is to explore the reasonable use of Doppler radar data assimilation to correct the initial and lateral boundary conditions and chose the optimal data assimilation mode. We will investigate the optimal B matrix systematically in further study by using 3-DVar, 4-DVar, EnKF and ETKF-3DVAR.

References: Liu, J., Bray, M., Han, D. Exploring the effect of data assimilation by WRF-3DVar for numerical rainfall prediction with different types of storm events, Hydrol. Process., 27, 3627-3640, doi: 10.1002/hyp.9488, 2013. Wang, H., Sun, J., Fan, S., Huang, X. Indirect assimilation of radar reflectivity with WRF 3D-Var and its impact on prediction of four summertime convective events, J. Appl. Meteorol. Clim., 52, 889-902, doi: 10.1175/JAMC-D-12-0120.1, 2013. Blni, G., Berre, L., Adamcsek, E. Comparison of static mesoscale background-error covariances estimated by three different ensemble data assimilation techniques, Q. J. Roy. Meteor. Soc., 141, 413-425. 2015.

Point 5: The description of the evaluation process is unclear and insufficient. No information (map illustration, data temporal analysis and coverage etc.) is presented concerning the rain gauges (Page 5, L. 6-7) used for evaluating the model results. No

information is presented concerning the method for pairing the model output and observations (e.g., nearest neighbor?). What do you mean by "areal rainfall observation at each rain gauge i" (Page 5, L. 15), since, the areal rainfall is calculated at the catchment scale using the observations from all 8 stations (Page 5, L. 6-7)? In overall, the terms "spatial" and "temporal" for computing the statistics CSI and RMSE are confusing. For example, to my understanding, spatial RMSE refers to the evaluation of the modeled 24-h rainfall considering all 8 stations, while temporal RMSE refers the evaluation of the basin-averaged rainfall using 24 model-observations pairs. However, both metrics consider the spatial dimension. Most studies in the literature apply the standard approach of domain-wide statistics (spatial dimension), using model-observation pairs of the examined variable (e.g., 1h or 24h rainfall) over all available stations, aggregated for certain time periods (temporal dimension). Thus, please consider revising the application of the statistics. Also, please consider computing more statistic metrics (e.g., POD, FAR etc.) to enhance the evaluation process. In the same direction, please consider evaluating the model results under different time intervals (e.g., 6h; 0, 6, 12, 18) and rain thresholds (e.g., >0.1, > 0.2 etc.). Finally, please provide information in the description of evaluation process concerning the construction and usage of Figures 4-9. For instance, do Figures 4-6 refer to the 24-h modeled and observed rainfall? Do Figures 7-9 refer to areal rainfall?

Reply: Thanks for the reviewer's suggestion. The spatial distribution of rain gauges has been added in Fig.2. The evaluation process is revised concerning the rain gauges used for evaluating the model results and the method for pairing the model output and observations. The first paragraph in section 5.2.1 is revised as a whole:

[revised manuscript text omitted]

In addition, we really think carefully about the reviewer's suggestion that more statistic metrics (e.g., POD, FAR etc.) should be considered. However, according to our previous studies (Tian et al, 2017), the simulations high CSI always have high POD and low FAR. In terms of evaluating the rainfall simulations in spatial and temporal dimensions, the effects of the three indices are similar. More indices may make the manuscript complex and reader puzzled. The CSI can be considered as a comprehensive description of accuracy. That is why we use CSI. More information can be found in our previous studies.

References: Tian, J., Liu, J., Yan, D., Li, C., Yu, F. Numerical rainfall simulation with different spatial and temporal evenness by using a WRF multiphysics ensemble, Nat. Hazards Earth Syst. Sci., 17, 563-579, doi: 10.5194/nhess-17-563-2017, 2017.

Though the time window of the storm events is chosen as 24 h, the rainfall is concentrated mostly in only a few hours. If the time step is set as 6 h or more, the evaluation will be difficult. One hour may be the most suitable time step to evaluate the rainfall

simulation finely. If the reviewer thinks that the results in different time intervals are necessary to add in the manuscript, we will supply the description in next round of modification. If possible, we suggest to evaluate the maximum 3-h, 6-h and 12-h rainfall simulations using RE rather than evaluate the model results under different time intervals with CSI or other indices.

Figure captions are revised to make the statements clear:

"Figure 4: Spatial distribution of the simulated 24 h rainfall accumulations with nine data assimilation modes for Event I. Figure 5. Spatial distribution of the simulated 24 h rainfall accumulations with nine data assimilation modes for Event II. Figure 6. Spatial distribution of the simulated 24 h rainfall accumulations with nine data assimilation modes for Event III. Figure 7: Time series bars of observed and simulated areal rainfall with nine data assimilation modes and the rainfall observation for Event I. Figure 8: Time series bars of observed and simulated areal rainfall with nine data assimilation modes and the rainfall observation for Event II. Figure 9: Time series bars of observed and simulated areal rainfall with nine data assimilation modes and the rainfall observation for Event III."

Point 6: 2/3 typhoon events affected the study area as tropical cyclones and had a limited impact in the study area. This fact does not support the aim of the study, which focus of typhoon rainfall simulations. I suggest including more high-impact typhoon rainfall events in the study. Also, please refer in more detail to the impacts on properties, people etc. in the study catchment, as well as to the flooding mechanisms (fluvial?) in the area. This will assist the results interpretation in terms of natural hazard analysis.

Reply: Much thanks for your suggestion that can make the manuscript closer to the aims and scope of the NHESS. In order to investigate the radar data assimilation effects on rainfall simulation, different kinds of rainfall processes caused by different stages of the typhoons. Rainfall storm event II occurs after the typhoon passes Meixi catchment

and the effects of the Hagibis have weakened. The spatial and temporal distribution of the rainfall is uneven. Event I and III both occur when the typhoons are close to Meixi catchment. Event I has relatively even spatial-temporal distribution of the rainfall, while event III is extreme rainfall. The most destructive flood caused by storm event III leads to many casualties and huge economic losses. The supplementary of disaster situation is added in the revised manuscript. The section 2 is revised as shown in Line 2-15, Page 4:

"The Meixi catchment lies in east-central of Fujian province with subtropical monsoon climate (Fig.1). The drainage area is 956 km2 and the average annual rainfall is approximately 1560 mm. There are 8 rain gauges and hydrologic station (Fig.2). In order to investigate the radar data assimilation effects on rainfall simulation, different kinds of rainfall processes caused by different stages of the typhoons are chosen in Meixi catchment. Saola forms on July 28, 2012 while lands Fuding, Fujian until August 3. With moving inland slowly, Saola weakens into a tropical storm at Jiangxi. Although event I occurs during the movement of Saola to Meixi catchment, the accumulated 24-h rainfall is only 84 mm. Hagibis lands Shantou, Guangdong on June 15, 2014 and then moves toward north with a fast-moving speed. Fortunately, Hagibis weakens into a tropical depression quickly during moving to northeastern Fujian on June 17. Event II occurs after the typhoon passes Meixi catchment and the accumulated 24-h rainfall is only 66 mm. Nepartak reaches Fujian on July 9 and strengthens at Putian. Then Nepartak moves towards the northwest with a fast-moving speed and event III occurs when Nepartak is close to Meixi catchment. During the period, Nepartak reaches its peak intensity. The 24 h accumulated rainfall is 242 mm and peak flow reaches 4710 m3/s in Meixi catchment. The most destructive flood causes water and power cut-off in 11 villages and towns. Official figures stand at 74 dead and 15 missing from the flood, which also causes a direct economic loss of 5.234 billion yuan. Accurate rainfall forecasts appear to be particularly important for Meixi catchment. Three rainfall storms are shown in Table 1."

Point 7: Please provide evidence on how assimilating radial velocity and radar reflectivity affect the WRF model's initial and boundary conditions (ICBC), and performance during the conducted numerical experiments. For example, you could compare the ICBC wind field and water vapor transportation between the experiments. This is important to support the interpretation of the results.

Reply: We agree with the suggestion that can support the interpretation of the results. Each rainfall mainly concentrates in a short period, so the wind field and water vapor transportation increment for different modes at the rainfall concentrating time are used to show how assimilating radial velocity and radar reflectivity affect the WRF model's initial and boundary conditions (Fig. 11-13). The shadows in Fig. 11-13 mean that water vapor transportation in analysis field is more than in background field. The darker the shadow in the figures, the more water vapor transportation increment.

[revised manuscript text omitted]

Minor comments

Point 8: Page 2, L. 22-27, 29-31 and Page 3, L. 1-2: Please refer to models and data assimilation schemes used.

Reply: The models and data assimilation schemes used in these studies are added in Line 22-32, Page 2 and Line 1-6, Page 3:

"Wang et al. (2013) tested the four-dimensional variational data assimilation (4-DVar) system by simulating a midlatitude squall-line case in the U.S. Great Plains, and the results indicated that radar data assimilation was able to improve rainfall forecasts from the WRF model at the convective scale. Liu et al. (2013) selected 4 storm events in a small catchment (135.2 km2) located in southwest England to explore the effect of data assimilation for rainfall forecasts based on WRF model, and assimilating radar reflectivity by 3-DVar model can significantly improve the forecasting accuracy for the events with one-dimensional evenness in either space or time. By using the WRF model and Advanced Regional Prediction System (ARPS) 3-Dvar, Hou et al. (2015) improved the short-term forecast skill up to 9 hours by assimilating radar data in southern China.

Most studies focus on the assimilation algorithm and data selection. However, consistent conclusions have not been obtained for the option of radar reflectivity and radial velocity, and few studies pay attention on the time interval setting of data assimilation. Based on the WRF and 3-DVar model, Tian et al. (2017b) found that radar reflectivity assimilation led to better rainfall simulation than radial velocity assimilation with the time interval of 6 h. Maiello et al. (2014) assimilated both radar reflectivity and radial velocity by 3-DVar model with 3 h assimilation cycle to improve the WRF high resolution initial condition, and the rainfall forecast became more accurate for several experiments in the urban area of Rome. Bauer et al. (2015) used the WRF model in combination with 3-DVar scheme to estimate the rainfall simulation, and the results showed that radar data assimilation significantly improved the rainfall simulation by a 1-hour Rapid-Update Cycle with the high resolution of 3 km in Germany."

Point 9: Page 4, Section 2.2: The description could be improved in terms of English and details provided.

Reply: The Section 2.2 (section 3.1.2 in the revised manuscript) are rewritten: "The fundamental of 3-DVar data assimilation is to produce an optimal estimate of the true atmospheric state by the iterative solution of a prescribed cost function (Ide et al., 1997): where x is the vector of the analysis, xb is the vector of first guess or background, y is the vector of the model-derived observation that is transformed from x by the observation operator H, i.e., y=H(x), and y0 is the vector of the observation. B is the background error covariance matrix, and R is the observational and representative error covariance matrix. Equation (1) shows that the 3-DVar is based on a multivariate incremental formulation. Velocity potential, total water mixing ratio, unbalanced pressure and stream function are all preconditioned control variables. Radial velocity has already been derived into component winds that are the same as the analysis variables, hence radial velocity can be assimilated directly by Eq. (1). However, radar reflectivity assimilation needs additional forward operator that associates the model hydrometeors with the radar reflectivity. Due to the wide applicability, the matrix of CV3 is adopted in this study to simplify the data assimilation procedure (Meng and Zhang, 2008)."

Point 10: Please refer to what is being shown in Figures 4-9 (24-h rainfall? Areal rainfall? See comment c) in Methodology above).

Reply: Figure captions are revised as follows:

Figure 4: Spatial distribution of the simulated 24 h rainfall accumulations with nine data assimilation modes for Event I. Figure 5. Spatial distribution of the simulated 24 h rainfall accumulations with nine data assimilation modes for Event II. Figure 6. Spatial distribution of the simulated 24 h rainfall accumulations with nine data assimilation modes for Event III. Figure 7: Time series bars of observed and simulated areal rainfall with nine data assimilation modes and the rainfall observation for Event I. Figure 8: Time series bars of observed and simulated areal rainfall with nine data assimilation modes and the rainfall observation for Event II. Figure 9: Time series bars of observed and simulated areal rainfall with nine data assimilation modes and the rainfall observation for Event III.

Point 11: Please enhance the resolution of Figures 7-9.

Reply: The resolution of Figures 7-9 is improved in the revised manuscript.

Point 12: Some examples where the English could be improved. Title: Please replace "simulation" by "simulations". Page 2, L. 2 and 5: Please replace "system" by "systems". Page 2, L. 10: Please replace "by" by "using the". Page 2, L. 14: Please replace "WRF-LTNGA" by "the WRF-LTNGA scheme". Page 2, L. 16-17: Please move "for hydrological applications" to the previous sentence ("...into the WRF model for hydrological applications"). Page 2, L. 19-20: Please rephrase. Page 2, L. 21: Please replace "the" by "their". Page 2, L. 28: Please remove "the". Page 3, L. 5: Please change to "caused by the interaction of typhoons and the complex terrain". Page 3, L. 6-7: Please rephrase. Page 3, L. 11: Please rephrase "flood disasters have attacked...". Page 3, L. 23: Please replace "can be" by "is". Page 3, L. 28: Please add "a" ("...has a significant effect..."). Page 5, L. 13-14: Please rephrase "... and 24 for N is the ...". Page 6, L. 1-2: Please use past tense.

Reply: The gramma mistakes and spelling errors are checked carefully. The errors mentioned by the reviewer are all revised accordingly.

[Figure]

[Figure]

Figure 11. Wind field and water vapor transportation increment (850hPa) for Event I at 12:00 on August 3, 2012

**Fig. 1.** Figure 11

Figure 12. Wind field and water vapor transportation increment (850hPa) for Event II at 18:00 on June 18, 2014

**Fig. 2.** Figure 12

[Figure]

Figure 13.Wind field and water vapor transportation increment (850hPa) for Event III at 6:00 on July 9,
2016

**Fig. 3.** Figure 13

$$J(\mathbf{x}) = \frac{1}{2}\left(\mathbf{x} - \mathbf{x}^b\right)^T \mathbf{B}^{-1}\left(\mathbf{x} - \mathbf{x}^b\right) + \frac{1}{2}\left(\mathbf{y} - \mathbf{y}^0\right)^T \mathbf{R}^{-1}\left(\mathbf{y} - \mathbf{y}^0\right)$$

**Fig. 4.** Equation

---

## Referee Report (RR1)

**2nd Round** Review of "Typhoon rainstorm simulation with radar data assimilation in southeast coast of China" by Tian J., Liu R., Ding L., Guo L. and Zhang B. (Manuscript ID: NHESS-2020-146)

I would like to thank the authors for addressing extensively all the review comments. The manuscript has been adequately improved. However, I still have some concerns that can be found below:

1) The computation of CSI and m-RMSE in terms of spatial and temporal distribution is still confusing for the reader. As I mentioned in the first review round, it seems that spatial m-RMSE refers to the evaluation of the modeled 24-h rainfall considering all 8 stations, while temporal m-RMSE refers the evaluation of the basin-averaged (areal) rainfall using 24 model-observations averaged over all 8 stations. Is that correct? If yes, the both metrics consider the spatial dimension. Thus, the spatial-temporal discretization has no point. Similar conclusions can be drawn for CSI.
Furthermore, the term "areal" is still unclear. Do you refer to the average of the modeled rainfall over the 8 rain gauges or to the basin-averaged modelled rainfall? In Figures 7-9, the rainfall observations at each hour are averaged over the 8 stations? To the end, do you compare the average of the modeled and observed rainfall over the 8 rain gauges or the basin-averaged modelled and observed rainfall? In the latter case, how do you compute the basin-averaged observed rainfall since you have point observations?
I suggest finding a more clear and robust way to describe the different metrics and its applications.

2) I still do not understand the phrase "Considering the application effect in southeast coast of China". I suggest keeping the justification of the physics options selection based on previous studies.

3) English grammar and style still need further improvements (e.g., use of past tense in the description of the studied events in Section 2).

---

## Author Response (AR2)

**Typhoon rainstorm simulations with radar data assimilation in southeast coast of China**

**Reply to Referee #2**

**Comments:**

**Point 1:** The computation of CSI and m-RMSE in terms of spatial and temporal distribution is still confusing for the reader. As I mentioned in the first review round, it seems that spatial m-RMSE refers to the evaluation of the modeled 24-h rainfall considering all 8 stations, while temporal m-RMSE refers the evaluation of the basin-averaged (areal) rainfall using 24 model-observations averaged over all 8 stations. Is that correct? If yes, the both metrics consider the spatial dimension. Thus, the spatial-temporal discretization has no point. Similar conclusions can be drawn for CSI.

Furthermore, the term "areal" is still unclear. Do you refer to the average of the modeled rainfall over the 8 rain gauges or to the basin-averaged modelled rainfall? In Figures 7-9, the rainfall observations at each hour are averaged over the 8 stations? To the end, do you compare the average of the modeled and observed rainfall over the 8 rain gauges or the basin-averaged modelled and observed rainfall? In the latter case, how do you compute the basin-averaged observed rainfall since you have point observations?

I suggest finding a more clear and robust way to describe the different metrics and its applications.

**Reply:** In this study, the observation of areal rainfall in Meixi catchment is averaged by the 8 stations with Thiessen polygon method, while the simulation of areal rainfall is averaged from all grids of the WRF model inside the Meixi catchment.

As mentioned in the manuscript, $P'_j$ and $P_j$ refer to the simulation and observation of 24-h accumulated rainfall at rain gauge $j$ for spatial m-RMSE. That is to say, the spatial dimension evaluation is not affected by time dimension. For temporal dimension evaluation, $P'_j$ and $P_j$ are the simulation and observation of areal rainfall at each time $j$, respectively. That is to say, the temporal dimension evaluation is not affected by the spatial differences of rainfall.

For *CSI, $NA_i$, $NB_i$, $NC_i$* at each time step $i$ ($i$=1 h) are calculated by comparing the rainfall observation with simulation extracted at 8 rain gauge locations in spatial dimension, and then the values of $NA_i$, $NB_i$, $NC_i$ at all time steps are averaged to produce the final verification results. Therefore, $N$ refers to the total time steps, which is 24. For temporal dimension evaluation, $NA_i$, $NB_i$, $NC_i$ are first calculated using the time series data of simulations and observations at each rain gauge $i$ ($i$= 1), then the averaged index values of all rain gauges are regarded as the final verification results. Thus instead of the simulation time steps, $N$ represents the total number of the rainfall gauges, which is 8 for temporal

dimension evaluation. That is to say, the temporal CSI is the average index values of all rain gauges. The calculation method can also be found in the following reference:

Liu J., Bray M., Han D. Sensitivity of the Weather Research and Forecasting (WRF) model to downscaling ratios and storm types in rainfall simulation, Hydrol. Process., 26, 3012-3031, doi: 10.1002/hyp.8247, 2012.

Therefore, the rainfall observations at each hour are averaged over the 8 stations by the Thiessen polygon method in Figures 7-9. We compare the average of the modeled and observed rainfall over the simulated rainfall averaged from all grids of the WRF model inside the Meixi catchment and the observed rainfall averaged by the 8 stations with Thiessen polygon method.

In order to make the "areal" clear, the following sentences are added in Line 27-29, Page 6:

*"In this study, the observation of areal rainfall in Meixi catchment is averaged by the 8 stations with Thiessen polygon method (Sivapalan and Blöschl, 1998), while the simulation of areal rainfall is averaged from all grids of the WRF model inside the Meixi catchment."*

The sentences in Line 2, Page 7 are revised as:

*"where P' is the simulation of 24-h accumulated areal rainfall, and P is the observation of 24-h accumulated areal rainfall."*

The description of metrics in Line 9-15, Page 7 are revised as:

*"$NA_i$, $NB_i$, $NC_i$ at each time step i (i=1 h) are calculated by comparing the rainfall observation with simulation extracted at 8 rain gauge locations, and then the values of $NA_i$, $NB_i$, $NC_i$ at all time steps are averaged to produce the final verification results. Therefore, N refers to the total time steps (N=24). For temporal dimension evaluation, $NA_i$, $NB_i$, $NC_i$ are first calculated using the time series data of simulations and observations at each rain gauge i (i= 1), then the averaged index values of all rain gauges are regarded as the final verification results. Thus instead of the simulation time steps, N represents the total number of the rainfall gauges (N=8) for temporal dimension evaluation."*

The sentences in Line 19, Page 7 are revised as:

*"$P'_j$ and $P_j$ are the simulation and observation of areal rainfall at each time j, respectively."*

**Point 2:** I still do not understand the phrase "Considering the application effect in southeast coast of China". I suggest keeping the justification of the physics options selection based on previous studies.

**Reply:** Thanks for the reviewer's suggestion. The sentence has been revised as:

*"According to the previous studies on physics options selection."*

**Point 3:** English grammar and style still need further improvements (e.g., use of past tense in the

description of the studied events in Section 2).

**Reply:** The gramma mistakes are checked carefully and English style is unified in the revised manuscript.